

# Long-term eddy modulation inhibited the meridional asymmetry of halocline in the Beaufort Gyre

Jinling Lu[1,2], Ling Du[1,2], Shuhao Tao[1,2]

[1] Frontier Science Center for Deep Ocean Multispheres and Earth System (FDOMES) and Physical Oceanography Laboratory, Ocean University of China, Qingdao, 266100, China
[2] College of Oceanic and Atmospheric Sciences, Ocean University of China, Qingdao, 266100, China

*Correspondence to:* Ling Du (duling@ouc.edu.cn)

**Abstract.** Under the background of wind forcing change along with Arctic sea ice retreat, the mesoscale processes undergoing distinct variation in Beaufort Gyre (BG) region are more and more significant to oceanic transport and energic cascade, and then these changes put oceanic stratification into a new state. Here the varying eddies and eddy kinetic energy (EKE) in the central Canada Basin (CB) and Chukchi–Beaufort continental slope are obtained based on mooring observations (2003–2018), altimetry measurements (1993–2019) and reanalysis data (1980–2021). In this paper, the variability of halocline in BG representing adjustment of stratification in the upper layer is shown so as to analyze how it occurs under significantly changing mesoscale processes. We find that the halocline depth has deepened by ~40 m while that in the north has deepened by ~70 m in the in the last nearly two decades by multiple data sets. The halocline depth lifting to the north initially was shifted to a final nearly symmetric structure. Eddy strength and Eddy induced low salinity water transportations have been continuously increasing toward the central basin at the mean time the halocline depth and strength among the southern and northern parts in the basin have reached a nearly identical and stable regime. It is clearly clarified that the long-term dynamical eddy modulation through eddy fluxes facilitating the freshwater redistribution inhibited the meridional asymmetry of halocline of the BG. Further research into high-resolution observations and data simulations can helps us to better understand the eddy modulation processes and its influence on large-scale circulation.

## 1 Introduction

Global temperatures have continued to rise since 1970s. The Arctic Ocean as the focal point of climate change research is the region with the most dramatic global surface temperature warming (Huang et al., 2018), with the warming range as high as 1.2 °C /10a, more than twice the global average warming range, which is called "Arctic amplification" phenomenon (Serreze and Barry, 2011). These variations not only affect the upper ocean circulation, but also expose the Arctic atmosphere–ice–sea system to rapid changes (Timmermans and Marshall, 2020). In this context, with summer sea ice declining in the Arctic (Stroeve et al., 2007, 2014; Niederdrenk and Notz, 2018) shown by satellite derived data, the existence of more freshwater in the upper layer makes local stratification alter and results in the redistribution of water masses. Meanwhile, the emergence of



broader areas of open water in the Canada Basin (CB) leading to more active ocean–atmosphere interaction and more susceptible to atmospheric forcing have attracted more and more attention to the mesoscale processes.

The Beaufort Gyre (BG) located in the CB, a large-scale wind-driven anticyclonic circulation feature, storing a substantial amount of freshwater in the CB (Proshutinsky et al., 2009, 2019), is accompanied by prevalent mesoscale eddies (Doddridge et al., 2019; Manucharyan and Spall, 2016; Zhao and Timmermans 2015; Zhao et al. 2016). The halocline in the CB, a thick

layer with a double peak of stratification, is considered to be an insulating "density barrier" between the surface mixed layer and Atlantic water layer underneath (Bourgain and Gascard, 2011). The asymmetric stratification of the BG and halocline vertical structure are payed attention in the recent researches (Kenigson et al., 2021; Zhang et al., 2023). The gyre is highly asymmetric associated with surface forcing and topography with isohalines steeper in the south and east compared with those in the north and west (Zhang et al., 2023). The increase of isopycnal slope with depth can be attributable to the eddy-induced

streamfunction (Kenigson et al., 2021). Besides, the freshwater content (FWC) accumulated by Ekman convergence has increased between 2003 and 2008 and remained relatively constant between 2008 and 2012 (Timmermans and Toole, 2023). Likewise, observations indicated that Pacific Winter Water (PWW) layer has generally deepened during 2004–2018 while the layer thickness has increased (Kenigson et al., 2021), which was identified an isopycnals deepening by 70 m during 2004–2011 (Zhong et al., 2018), suggesting a spin-up of the gyre. Previous works about eddies in the CB or the Arctic Ocean were

mostly based on satellite (e.g., Kozlov et al., 2019; Kubryakov et al., 2021, Raj et al., 2016), in situ hydrographic data (e.g., Fer et al., 2018; Timmermans et al., 2008; Zhao et al., 2014, 2016; Zhao and Timmermans, 2015), high eddy-resolving simulations (e.g., Reagan et al., 2020; Wang et al., 2020) and etc. Eddy activity, a common feature in the halocline of the BG, is also focused by many past studies. Moreover, the kinetic energy in the halocline of BG was mainly dominated by mesoscale eddy activities (Zhao et al., 2016, 2018). Eddies are distributed at different depths in the Arctic Ocean and mainly concentrated

at subsurface (Zhao et al., 2014) even they may extend to thousands of meters in depth. The depth of maximum value is generally found about 70–110 m in the halocline (Wang et al., 2020). Based on 127 eddies observed at drifting sea ice stations, Manley and Hunkins (1985) found that the eddy kinetic energy (EKE) accounted for about one-third of the total kinetic energy (TKE) of the upper 200 m in the CB. From the perspective of the horizontal pattern, EKE derived by satellite is also higher along main boundary currents and continental shelves in the Arctic Ocean (Timmermans and Marshall 2020). Zhao et al. (2016)

kept Ice Tethered Profiler (ITP) measurements for temperature, salinity and current between 2005 and 2015 to survey the changes of eddy field in the CB. They found that eddies were mostly distributed in the western and southern parts of the CB. As was showed that the number of eddies in the lower halocline doubled from 2005–2012 to 2013–2014 (Zhao at al., 2016) with the past increasing of FWC, the gyre areas and strength (Regan et al., 2019; Timmermans and Toole, 2023; Zhang et al., 2016). The response of TKE and EKE to the spin-up of the gyre during 2003–2007 in particular showed that EKE at subsurface

has generally strengthened (Regan et al., 2020). It was also demonstrated by a recent research that with wind energy input increasing into BG due to significant loss of sea ice after 2007 eddy activities would also be more active (Armitage et al., 2020).





Mesoscale eddies can transmit momentum, heat, water masses and chemical compositions, not only contributing to atmospheric circulation, mass distribution and marine biology, but also playing an important role in global ocean heat balance (Chelton et al., 2007). Eddies are not only exhibiting unprecedented changes but also playing a crucial role in the Ekman-driven BG stability in the context of sea ice loss (Manucharyan et al., 2016). They can balance atmosphere–ocean and ice–sea stress input, gradually weaken the slope of isopycnals and geostrophic currents and counteract the accumulation of FWC driven by Ekman pumping through dissipating available potential energy (APE). The eddy activity as a key physical process affects the release and accumulation of freshwater and ultimately influences the formation of halocline (Manucharyan and Spall, 2016). Except that, the Ekman pumping and sea ice are also major factors affecting the dynamics of halocline. This balance between halocline and eddies is thought to occur on different time scales in realistic models (Doddridge et al., 2019; Manucharyan et al., 2017) and suggests a link between small-scale features and changes to the large-scale circulation.

However, with sea ice condition changing due to global warming, long-term variability of eddies in the central basin and basin boundary regions is still unsolved. Furthermore, according to the standpoint about possible gyre's stabilization in recent years (Proshutinsky et al., 2019; Zhang et al., 2016) the eddy modulation in the halocline of the BG on a long timescale is still unknown. Due to the influence of the measurement conditions, limited satellite observation, now continuous observation data of eddies in space and time is relatively scarce. Data coverage in space and time is yet to be improved (Zhao et al., 2016). The results of numerical simulation lack effective data to support, so researches on oceanic mesoscale eddies remain uncertain to some extent. Here we used multiple data sets containing moored, in situ and satellite altimetry observations, in comparison with reanalysis data, to quantify the strength of mesoscale processes by SLA and horizonal currents. The stationary eddies and EKE as well as the transformation of halocline across the basin are both pointed out to assess the low frequency variability of halocline in BG under significantly changing mesoscale eddies. Section 2 presents the details of data and methodology. Section 3 demonstrates the halocline variability especially on its meridional asymmetry in the BG region. And eddy distribution and interannual changes are discussed in Section 4. Section 5 explains the correlation of EKE and geostrophic currents as well significant eddy modulation in the halocline. Section 6 is summary and discussion in this paper.

## 2 Data and methods

### 2.1 Observations and ocean reanalysis data

In this paper we used multiple data sets including hydrographic observations, satellite altimetry and reanalysis data sets. The hydrographic data are in situ measurements from Conductivity Temperature Depth (CTD) and mooring data from McLane Moored Profilers (MMPs) at four moorings that are all deployed under the Beaufort Gyre Exploration Project (BGEP, http://www.whoi.edu/beaufortgyre/data). The reanalysis data sets used here mainly consists of World Ocean Atlas 2018 (WOA18) and Simple Ocean Data Assimilation (SODA, version 3.4.2).



Annual hydrographic survey through ship based CTD has been conducted in the BG region each year between August and
October. CTD data between 2004 and 2021 are used to mainly investigate spatio-temporal variability of oceanic stratification
across the CB. The positions of CTD instruments deployed is shown in Fig. 1a. Plus, to supplement long-term trends and
changing characteristics of halocline and to capture mesoscale eddies at representative stations in the CB, mooring data
deployed at four corners around the basin (Fig. 1b) between mid 2003 and mid 2018 above 500 m are also analyzed. Each
mooring system included a MMP that returned profiles of horizontal velocity, temperature, salinity, pressure and etc. A pair
of upgoing/downgoing profiles (separated by 6 hours) was returned every other day, data were processed to a vertical resolution
of 2 dbar. The shallowest moored measurement varies from about 50–90 m (depending on the mooring and sampling period)
to avoid collisions with ice keels and the deepest measurements are to 2000 m.

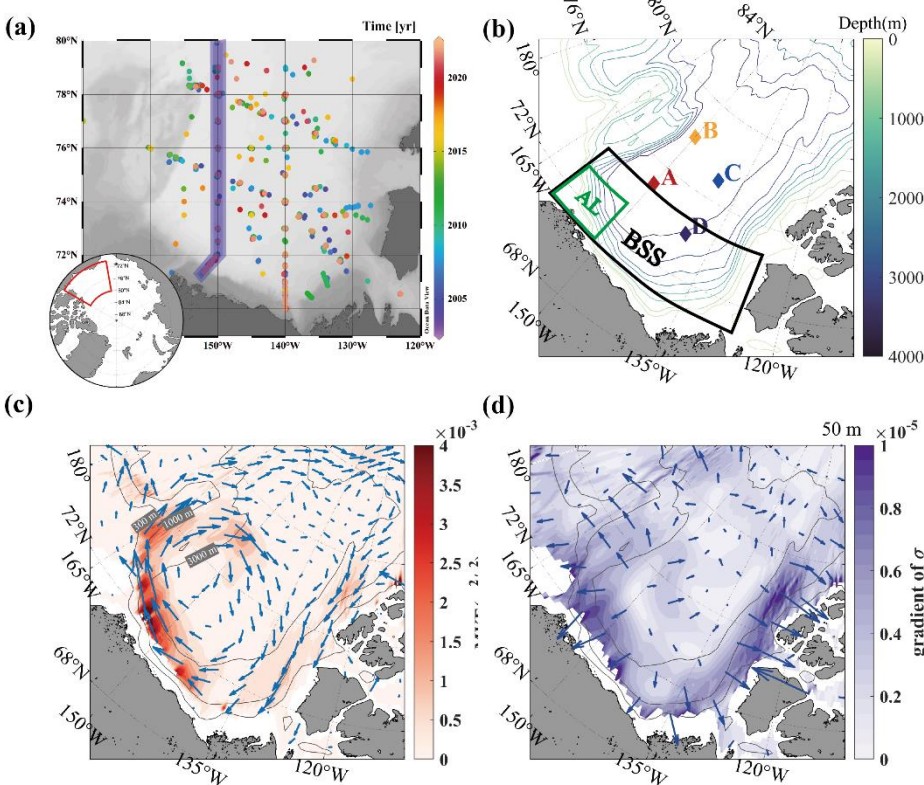

**Figure 1. (a) The positions of in situ sites of CTD measurement from BGEP in a certain months during 2004–2021. The purple bar**
**indicates an artificially selected meridional transect with a width of 36km mostly along 150°W but partially bent at the southwestern**
**continental slope in Beaufort Sea. (b) A map of the Canada Basin and the bathymetric contours upper than 4000 m isobath. Coloured**
**diamonds denote the locations of four BGEP moorings. The two chosen regions are shown by green (AL, Alaskan coast) and black**
**(BSS, Beaufort Sea slope) boxes respectively. (c) The distribution of mean kinetic energy at 50 m. Vectors denote the directions of**
**mean currents. Gray lines denote the 300 m, 1000 m and 3000 m bathymetry. (d) The distribution of horizontal gradient of potential**
**density (shading and vector) at 50 m. Vectors point in the directions of increasing potential density. The results of (c) and (d) are**
**calculated from the 2005–2017 WOA climatology.**



The SODA reanalysis data set is developed by the University of Maryland based on the Global Simple Ocean Data Assimilation System, which adopted in this paper is the 5 days averaged from 1980 to 2021, with a horizontal resolution of $1/2°×1/2°$ and vertically divided into 50 layers with unequal spacing. We obtained the gridded altimetry data (product identifier:
SEALEVEL_GLO_PHY_L4_REP_OBSERVATIONS_088_047) over the years in the 1993–2019 period from the Copernicus Marine Environmental Monitoring Service (CMEMS). This product consists of daily gridded maps of dynamic topography in ice-free regions that have been derived as a sum of mapped sea level anomalies (SLA) calculated from combined measurements by different satellites and mean dynamic topography (MDT).

## 2.2 Methods

For estimating EKE to assess the strength of eddy activities, we used ocean current data from and SODA and altimetry. Geostrophic velocities are calculated from sea level hight. The horizontal velocity is decomposed into annual mean velocity $(\bar{u}, \bar{v})$ and abnormal value $(u', v')$ (Penduff et al., 2004; Rieck et al., 2015, 2018; Regan et al., 2020):

$$u = \bar{u} + u', u = \bar{v} + v',$$
$$and\ then\ EKE = (u'^2 + v'^2)/2. \tag{1}$$

Note that the EKE in this paper is estimated by a low-frequency ''eddy'' which defined as a departure from a long-term temporal mean, with a period (it depends on temporal resolution of the data) of greater than 5 days or 1 day (Lucke et al., 2017). In addition, the vertical velocity shear $\partial U/\partial z$ can be related to the large-scale density field by the thermal wind relation (Meneghello et al., 2021)

$$\frac{\partial U}{\partial z} = \frac{g}{f_o \rho_o} \frac{\partial \rho}{\partial z} \vec{k} \times \nabla z_\rho = \frac{N^2}{f_o} \vec{k} \times \nabla z_\rho \tag{2}$$

where **U** is the horizontal current field, N is Brunt-Väisälä buoyancy frequency which represents oceanic stratification, $\nabla z_\rho = (-\frac{\partial \rho}{\partial x}/\frac{\partial \rho}{\partial z}, -\frac{\partial \rho}{\partial y}/\frac{\partial \rho}{\partial z})$ is the isopycnal slope, ρ is potential density of sea water, $\rho_o$ is the average density of seawater, g is the gravity acceleration, and z is depth. Developed by (3), the horizontal velocity field is calculated by the integration with depth from bottom to surface. The maps of horizontal velocity field (Fig. 1c) and density gradient (Fig. 1d) at 50 m in the CB are shown, the main circulation feature is clearly discerned and southwestern basin near continental slopes is the key region for
varying currents tending towards high EKE and instability.

For investigating the variation of halocline to understand shifting of oceanic stratification, we consider the depth of potential density surface $\sigma$=27.4 (25) kg • m$^{-3}$ to approximately represent the base (top) of the halocline (Timmermans et al., 2020). Based on the upper and lower boundary of halocline, APE is defined as the amount of potential energy in a stratified fluid available for mixing and conversion into kinetic energy (Huang 1998; Munk and Wunsch 1998) is following Eq. (3) (Polyakov
et al., 2018; Bertosio et al., 2022, partial modification):

$$APE = \int_{z_2}^{z_1} g[\rho(z) - \rho_{ref}]zdz, \tag{3}$$





where $z_1$ and $z_2$ represent the depth of halocline upper and lower boundary, and $\rho_{ref}$ is potential density at the base of the halocline.

Furthermore, for discerning the critical role of mesoscale eddies in balancing the halocline, we consider the eddy advection
velocity in the (y, z) plane can be defined from an eddy streamfunction $\psi^*$ as

$$v^* = -\psi_z^*, w^* = \psi_y^* \qquad (4)$$

and $\psi^*$ is represented as (Manucharyan et al., 2016; Manucharyan and Spall, 2016; Manucharyan and Isachsen, 2019)

$$\psi^* = \frac{\overline{V'S'}}{\overline{S_z}} = -\frac{\overline{w'S'}}{\overline{S_y}} \qquad (5)$$

where $\overline{V'S'}$ is the average meridional eddy salt flux and $\overline{S_z}$ is the average vertical salt gradient (Marshall and Radko, 2003).
Here bars and primes correspond to annul mean and perturbation variables. Due to buoyancy mainly controlled by salinity in Arctic, $\psi^*$ represents the cumulative effects of eddy thickness fluxes that arise from correlations between eddy velocities and eddy induced isopycnal displacements. Overall, when vertical salt gradient is generally negative in the CB a positive value of $\Psi^*$ indicates a southward (northward) transportation of high(low) salinity water and vice versa.

If eddy genesis is related to baroclinic instability, baroclinic growth rate ω is correlated with EKE. The baroclinic growth rate
ω can be estimated here by (Simth, 2007)

$$\omega = f\sqrt{\frac{1}{6H}\int_H^0 \frac{dz}{R_i(z)}} \qquad (6)$$

where $R_i = N^2/[\left(\frac{\partial u}{\partial z}\right)^2 + \left(\frac{\partial v}{\partial z}\right)^2]$ is the Richardson number. We call the inverse of this quantity $\omega^{-1}$=T the "Eady timescale". The Eady timescale should be short where there is anomalously high EKE or weak stratification.

### 3 BG halocline variability

This section is aimed to investigate the spatio-temporal variability of halocline in the BG region, particularly its varying asymmetry inside is the main focus of this article. The halocline's depth, thickness and strength and vertical structure are detailedly analyzed below, all of which indicate its meridional asymmetry at the mean time.

### 3.1 Temporal variation of the halocline

Under the spin-up of the BG, isopycnals of the PWW layer in the cold halocline have deepened (Kenigson et al., 2021). We
have chosen the special isopycnal surface to characterize the top and base of the halocline. Figure 2a and b show the discontinuous variation of the halocline upper/lower boundary and thickness at four moorings from MMP. As a whole, the rangeability of the halocline top is much smaller than that of the halocline base, and the depth of the halocline upper/lower boundary at single mooring shows basically consistent trends, so we mainly focus on the variation of halocline base depth. But there are different characteristics of variation during 2003–2018 despite of void measurements in time. Finite results at mooring
C show that an increasing trend of depth and thickness of the halocline before 2008. Besides, other moorings provided results

 

over a longer term, which captured a deepening of the halocline base and an increasing of thickness over the years from 2003 to 2018. The thickness of halocline at mooring B located in the northwestern part of CB increased steadily by about 70 m, at the same time the depth of halocline base deepened by up to 70 m over the years 2003–2018. The thickness of halocline in the southern part of the basin (moorings A and D) both increased by about 30 m company with the halocline base deepening

approximately 40 m. It's worth noting that the depth of halocline has a stagnant phase even opposite development over the years between 2003–2007 and 2015–2018. Particularly, linear trends and mean values of the halocline depth and thickness in three periods (2003–2007; 2008–2014; 2015–2018) are computed (Table 1). A negative trend of halocline depth is clearly during 2008–2014 in the southern sites of the basin (moorings A and D), but the former and latter periods both mostly exhibit positive trends in halocline depth and thickness. The variation at the only northern site (mooring B) covering three periods

shows entirely different features, the halocline thickness reveals a negative trend in the third period (after 2015) that eventually remains a steady level while the depth of that still keeps on deepening. In final, the halocline thickness and depth at every site tends to be homogeneously distributed and the differences are obviously shrunken than before.

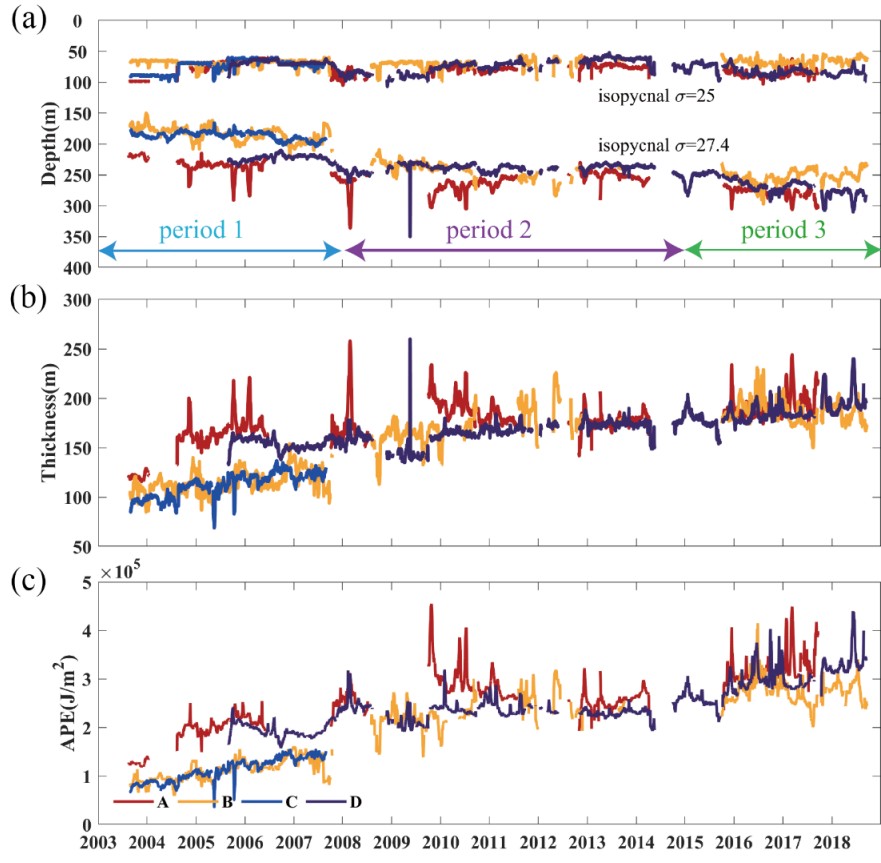

**Figure 2. Time series of (a) depth of isopycnals 25 kg/m³ (upper coloured lines) and 27.4kg/m³ (upper coloured lines) representing**
**the top and base of the halocline (b) halocline thickness between isopycnals 25 kg/m³ and 27.4 kg/m³ and (c) APE for moorings A, B, C and D during 2003–2018. Note that the abnormal values record eddies were existent at that time.**



APE, a good integral indicator of changes in overall halocline strength in the CB, is also computed here by Eq. (4). As is shown in Fig. 2c, the variation is similar with that of halocline thickness. Initially, APE revealed a striking difference between the northern (moorings B and C) and southern sites (moorings A and D) around the basin. The trend of APE showed a weak

decreasing after 2008 and then recovered to some extent at the southern moorings. In contrast, APE at the northern moorings kept on improving until 2014 and then the growth stagnates. The difference among moorings reduced in final, and APE all remained about $3 \times 10^5$ J/m$^2$, that was the maximum value over the years. We infer the variability of halocline and APE have a relationship with the BG spin-up and largest increasing of FWC during 2003–2007 (Giles et al., 2012; Krishfield et al., 2014; Timmermans and Toole, 2023). And then partial variables exist stagnant in the post spin-up term during 2008–2014 (Regan et

al., 2020).

**Table 1. Trends (whthin the brackets, unit: m/yr) and mean values (outside the brackets, unit: m) of the halocline's top, base and thickness in three periods for moorings A, B, C and D, respectively.**

| Moorings | | 2003–2007 | Periods 2008–2014 | 2015–2018 |
|---|---|---|---|---|
| | top | 75.4(–2.7) | 77.8(–2) | 86.2(–1.19) |
| A | base | 236.4(7.3) | 261.1(–4.49) | 278.1(7.91) |
| | thickness | 161.0(10.04) | 183.3(–2.48) | 191.9(9.10) |
| | top | 69.1(0.47) | 69.8(–1.98) | 66.8(–1.13) |
| B | base | 184.1(4.94) | 241.08(5.35) | 252.62(3.57) |
| | thickness | 115.0(4.46) | 171.28(7.33) | 185.82(–2.44) |
| | top | 74.26(–5.74) | | |
| C | base | 186.54(2.73) | | |
| | thickness | 112.29(8.47) | | |
| | top | 69.31(2.16) | 73.58(–3.92) | 81.52(3.09) |
| D | base | 223.69(0.4) | 239.44(–0.35) | 267.62(8.89) |
| | thickness | 154.37(–1.76) | 165.86(3.57) | 186.1(5.8) |



## 3.2 Changes of meridional asymmetry

The gyre located in the CB is marked by a pronounced asymmetry (Regan et al., 2019) with changing spatial distribution of the freshwater and ocean dynamic height. The isopycnal slope is steeper over the southern continental slope than that in the northern basin (Fig. 1d), almost in line with previous researches (Proshutinsky et al., 2019; Regan et al., 2019; Zhang et al., 2023). The former observations have revealed that isopycnals have deepened with different rates among the northwestern and northeastern parts in the basin during 2002–2016 (Zhong et al., 2019). According to section 3.1, we find the main differences of evolution only between northern and southern basin are obvious, which is not completely identical with previous findings. Therefore, we next turn to the inhomogeneous gridded in situ hydrographic data from the latest CTD observation so as to get a better understanding of overall asymmetric halocline across the basin. From the perspective of the horizontal maps in the three periods (Fig. 3) that are determined referring to the trends of halocline variables at the moorings, the spatial patterns of the halocline base and APE, implying the location and strength of the BG in the basin, both show evident changes. In the first period, maps of APE and halocline exhibit the same asymmetry and then there are an overall deepening of halocline as well as a gradual decreasing of spatial difference. The maximum of halocline depth is in the interior of the basin. At the mean time, APE in the latest period is much more remarkable than that in the first term along the continental slopes of Canadian Arctic Archipelago and Northwind Ridge where isopycnal gradient and baroclinic instability are significant as well as in the abyssal plain where the halocline base is deepest.

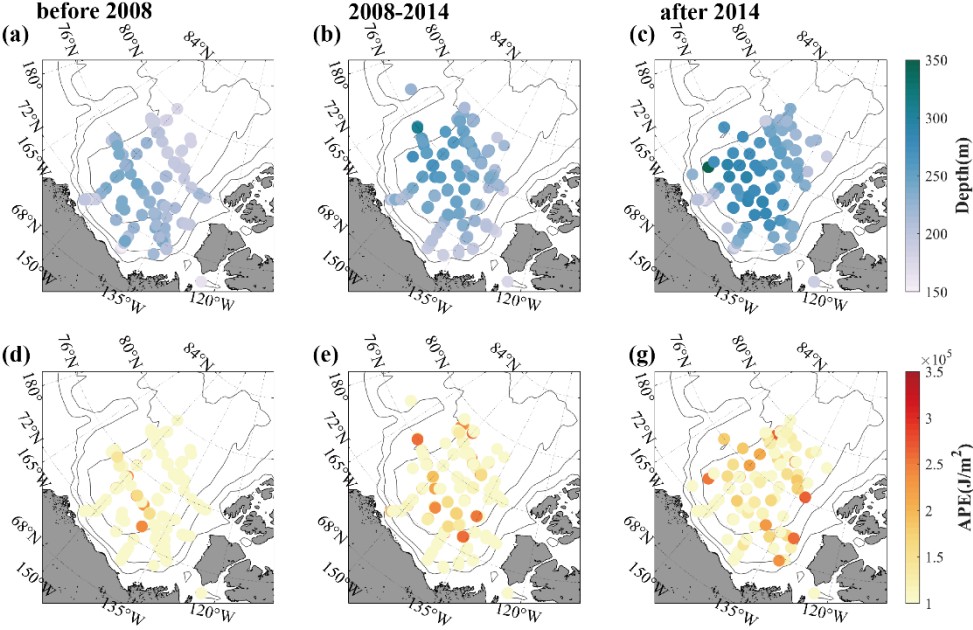

**Figure 3. (a-c) Horizontal distribution of depth of halocline base across the Canada Basin in 2004–2007 (before 2008), 2008–2014, and 2015–2021 (after 2014), respectively. (d-e) As the same with (a-c), but for APE in every period (integration between the top and base of halocline).**

 

In addition, the in situ hydrographic data are interpolated onto the regular grids to examine the varying vertical structures of halocline along the selected transect (Fig. 1a). Notably, the hydrographic structures along 150° W and 140° W sections are similar (Timmermans and Toole, 2023). Thus, we only select a representative north–south transect mainly along 150° W to analyze here. The vertical distribution of the isopycnal $\sigma$ = 27.4 kg·m$^{-3}$ surface show that it is shallowest ~ 200 m at the margins of the BG region and up to 80 m deeper in the interior BG in the later years (Fig. 4). Among the early, median and

later years shown, the vertical structures of isopycnals especially the lower boundary of the halocline reveal apparent changes between the marginal and interior gyre. From transects of potential density (Fig. 4), initially there was an distinct uplift of the halocline towards the north with the depth of halocline base in the south (~74° N) about 50 m lower than in the north (~77° N). The difference between the north and the south was narrowed with isopycnals generally deepening from the view of the average vertical structure during 2008–2014, and even the northern halocline is lower than southern district (the difference is

less than 10 m). In the third period (after 2014), the depth of the halocline has changed less in comparison with the previous periods, and the halocline is clearly meridionally symmetric shaped like a horizontal bowl, as if it has reached a state of equilibrium. As can be seen from the spatial maps and vertical structures of the halocline and APE, the characteristic of meridional asymmetry was gradually weakening in recent years. We infer there is possibly existing other physical process join in the variability and we are plan to discuss below.

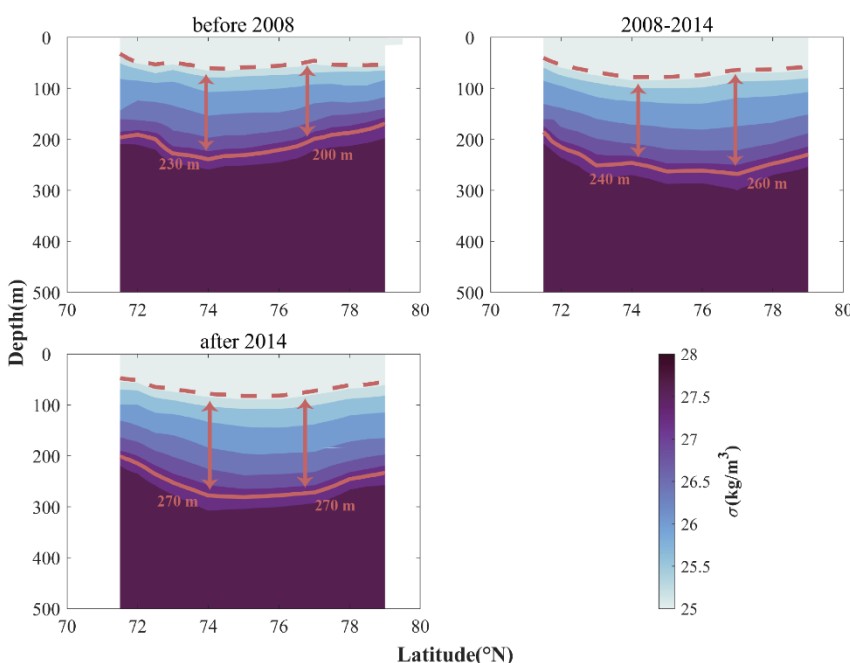


**Figure 4. Vertical transects of potential density using data from CTD measurements in 2004–2007 (before 2008), 2008–2014, and 2015–2021 (after 2014), respectively. The dashed (solid) lines indicate depth of $\sigma$ = 25 (27.4) kg·m$^{-3}$ representing the top (base) of halocline.**



## 4 Spatio-temporal variability of eddy activity

As was revealed by the former research, a regime shift of the BG occurred in 2007–08, with a spin-up phase of the gyre occurred from 2003 to 2007 and a stabilization after 2007 (Regan et al., 2020). The depth, strength, core location of halocline all imply the shift of the gyre and FWC variability. With BG spin-up and environmental conditions changing, mesoscale eddies are responding to dissipate extra energy input and influence the potential energy redistribution. In this section the spatio-temporal variability of eddies by eddy detection and EKE (a critical criterion to measure the strength of eddies) will be

discussed.

### 4.1 Eddy vertical distribution

We now outline how mesoscale eddies can be detected based on moored observations. When eddies occur locally, there are strong horizontal velocities accompanied by isopycnal displacements. As for anticyclonic (cyclonic) eddies, the isopycnals are convex (concave). By distinguishing the anormal speeds larger than 10 cm/s and the isopycnal displacements which are both

the criterion used in the past works (Timmermans et al., 2008; Zhao et al. 2014; Zhao and Timmermans, 2015), we counted the annual number of eddies in the upper layer (Fig. 5a). In all, there are 37, 40, 7 and 43eddies detected above 500 m at mooring A-D, respectively. They are mostly concentrated between the upper and lower halocline boundaries. As is the same with previous works (e.g., Zhao et al., 2014; Zhao and Timmermans, 2015), in the majority of instances, the abnormal temperature/salinity and convex isopycnal displacements in the eddy core are pervasive. The all cold-core eddies are accounted

for 61.4%. Most of these are anticyclones and only 3 eddies detected at mooring C are cyclones. The cold-core anticyclones are popular in the BG region due to oceanic stratification and large-scale dominated circulation. Furthermore, for the location of mooring C less controlled by the BG, characteristics of eddies there is different from others. Some of eddies are cyclone that are seldom discovered at other moorings. The existence of cyclones are related to frontal instability near 80° N that contributes the eddy formation (Manucharyan and Timmermans, 2013; Timmermans et al., 2008).

Additionally, comparing the vertical structures of EKE profiles in three periods at the moorings, we find that EKE changed significantly above halocline in the three periods (Fig. 5b). The EKE below halocline is relatively weaker than that in the upper layer, and its multiyear variation is much smaller. The vertical structures of EKE in the basin and its marginal seas can be classified into two types. The first type is that EKE is surface-intensified up to ~ 0.01 $m^2/s^2$ at surface and it decays with depth. The second one is bimodal with separate comparably high values at surface less than 50 m and subsurface approximately 90–

250 m between the upper and lower halocline boundaries. In the first period, EKE above the BG halocline remained at a relatively low level, and it has increased in the second period when the BG circulation appeared to be stabilizing (Zhang et al., 2016). The results from three moorings (all of them except mooring C are detected after 2008) show that EKE was strengthened to varying degrees, accompanied by a deepening of the halocline lower boundary. At the southwestern corner (A) of the basin,EKE increased in the second period and remained stable in the third period; northwestern (B) EKE strengthened in the second

period and weakened in the third period; southeastern (D) subsurface EKE didn't occur apparent growth until the third period.



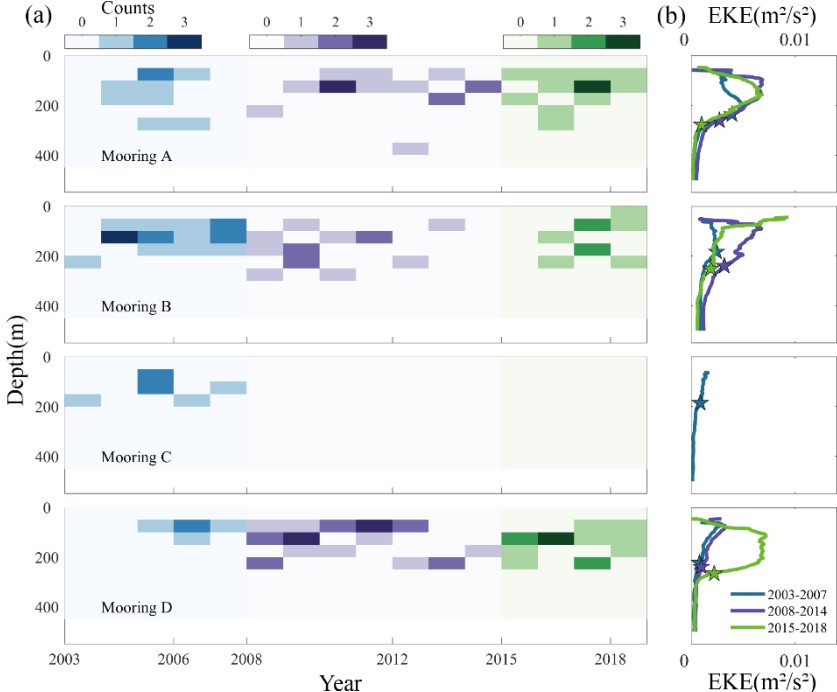

**Figure 5. (a) Hovmöller diagrams of depth against time showing annual single eddy counts in the upper layer at moorings A–D, respectively. Blue, purple and green shadings denote the spans of three periods. (b) Vertical profiles of mean EKE for four moorings over years in the three periods. Coloured stars indicate the depths of the halocline base in corresponding periods.**

## 4.2 Interannual EKE patterns

The BG region, a focal area for mesoscale phenomenon in the previous studies (Armitage et al., 2020; Regan et al., 2020; Zhao and Timmermans, 2015; Zhao et al., 2016), mainly consists of a southern narrow continental shelf close to the Alaska coast and a sizable deep basin. The Chukchi–Beaufort slope is the major sector for eddy generation by baroclinic instability (Spall et al., 2008) with a surface front approximately along 300 m isobath (Timmermans and Toole, 2023), and then eddies carrying pacific water propagate to the central BG by boundary current. Here we focus on this area to investigate the variety of EKE from a broad perspective used by satellite derived dynamic heights. Further, we seek for the main EKE patterns at surface in the three periods. As is shown (Fig. 6), the high value areas of EKE is mainly located along the continental slopes of the marginal CB especially the Alaska coast mostly between 1000 m and 3000 m isobaths. Indeed, energy is the strongest at the southwestern shelf break of CB near the Barrow Cape which can even reach more than $5 \times 10^{-3}$ m²/s² while it is even less than $1 \times 10^{-3}$ m²/s² in the interior basin. Notably, the horizontal pattern of EKE is not identical with that of MKE obtained by annual mean geostrophic current (not shown here). On the whole the area where EKE is strongest is closer to the inshore shelf sea side than the area where MKE is strongest. EKE in every term is all significantly enhanced comparing with that in the former term, and the strong EKE gradually developed from coasts to offshore and the central basin with time. For instance, from the interannual mean horizontal patterns the region with the strongest EKE was mostly concentrated at southern part of 72°N if





we only notice the section along 1000 m isobath before 2007 (1993–2007) and it extended to about 73°N in the next period.

Furthermore, the domain was even extended northward up to 74°N lying at the North Wind Ridge delineated by a long, clear

and curved ribbon in the latest period. We imply that eddy transportation contributes considerably to this development, which

still need more evidence.

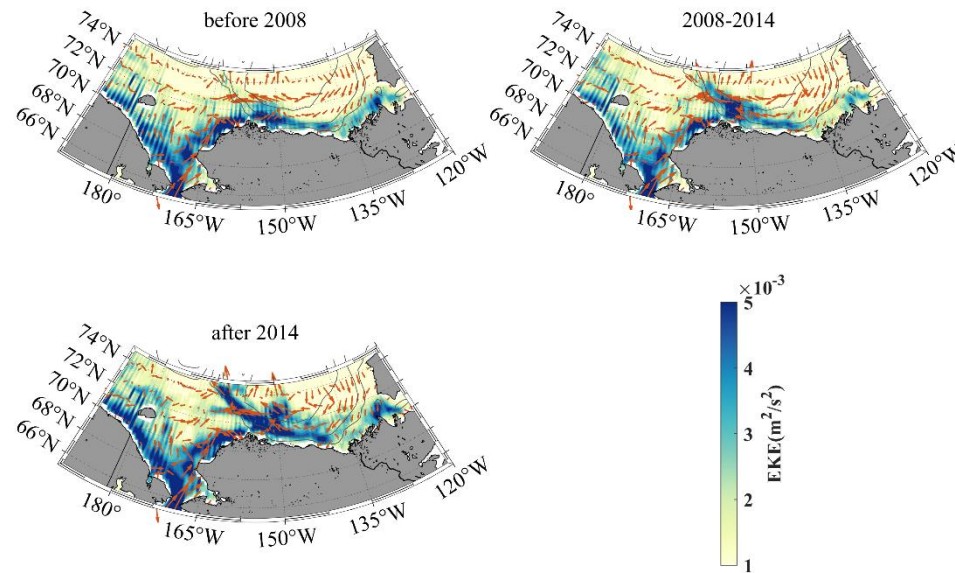

**Figure 6. Interannual mean maps of (shading) eddy kinetic energy (EKE) and (vector) geostrophic flows at surface in 1993–2007**
**(before 2008), 2008–2014 and 2015–2019 (after 2014), respectively.**

### 4.3 Long-term evolution

We confirmed the dates of existing eddies and annual number of warm or cold core eddies over 500 m through moored

observations. The interannual variations in days of recording mesoscale eddies and the number of eddies are very similar at

moorings A and B, and several respective peaks are predominant (Fig. 7, days of effective observations exceed 200 days in

most eddy-rich years) in the three periods of revealing critical halocline changes discussed in section 3. In every dominating

period of halocline variability there is one strong eddy-rich year coming to light under observation. Among them, 2005, 2010

and 2017 for mooring A are eddy-rich years; for mooring B, 2005, 2009 and 2018 (144 days record valid observations) are

eddy-rich years. We can see that the days of eddy activities demonstrate considerable interannual fluctuations. It is speculated

that the eddy genesis may be related to the accumulation and release of APE in the BG region and the transmeridional

movements of BG, which modulates the vertical structure of internal halocline. Plus, the year with abundant mesoscale eddies

at the northern site is more lagging about 1 year than that in the south latterly during medium term, which is affected by eddy

transportation and spatial inhomogeneity. Therefore, in the latest term specially after 2007 enhanced eddy activity is





noteworthy. Meanwhile, the amplitude of eddy activities at mooring D is obviously noticeable, although this site is far deviated
from the Chukchi–Beaufort continental slope, which is the key area for eddy generation (Kubryakov et al., 2021; Manucharyan
and Isachsen, 2019; Zhao et al., 2014). , The in situ measurement at mooring D captured a large number of mesoscale eddies,
with a deceasing trend of number during medium term in line with the 2003–14 northwestern movement of the BG center
(Regan et al., 2019). From 2017 onwards, the BG retreated from Mendeleev Ridge to the east (Moore et al. 2018) accompanied
by elevated eddy activities.

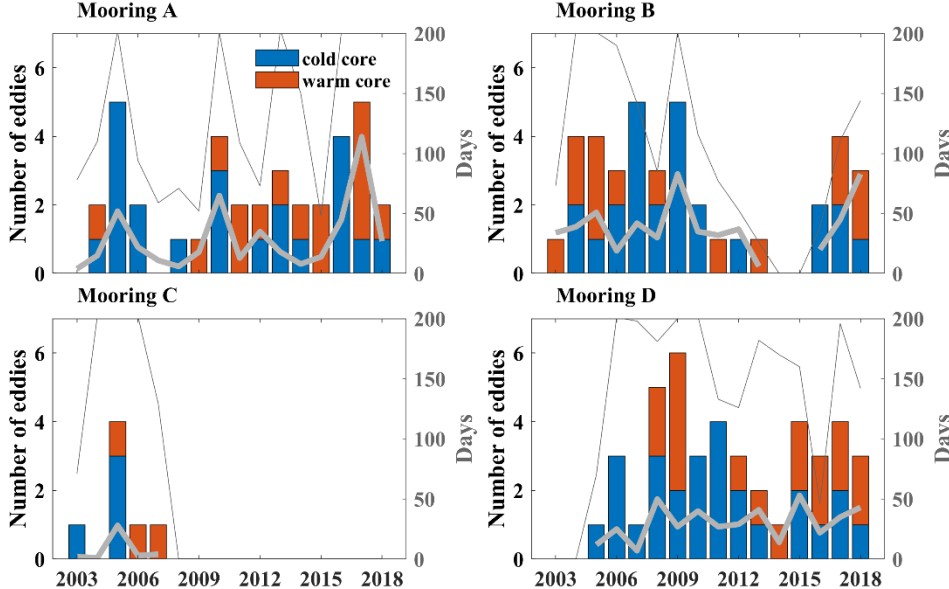


**Figure 7. Interannual evolution of the days of existing eddies (thick grey line) and number of eddies (bar) for four moorings. The
blue and red bars indicate the counts of cold-core and warm-core eddies. Thin grey lines signify the days of recording valid
observation in every year.**

Nowadays the seasonality of EKE in the Arctic clearly now that it is generally maximal in late summer or autumn and minimum
in spring or winter (Wang et al., 2020; Manucharyan and Thompson, 2022), which is similar with other global regions (Rieck
et al., 2015; Jia et al., 2011). Seasonal cycles of EKE in the central basin and basin boundary regions are distinct (Fig. 8b).
However, the researches about long-term EKE evolution are still fewer. The Alaska coast and the Chukchi–Beaufort Slope is
the key area of varying EKE. Moreover, we used finite data sets derived from SODA reanalysis, altimetry and moored
observations to explore the long-term variability of EKE between the central basin and continental slope. We selected a western
point of the Alaska coast called AL region here near the Barrow Cape (Fig. 1b) and the BG region representing the central
basin determined from the positions of four moorings. Here we think results of every mooring from MMPs are equally to
character the main features of mesoscale processes in the BG region, so the results above the halocline base of different
moorings are vertically averaged with depth so as to obtain a longer continuous change of EKE over the years between 2003
and 2018. The annual mean time series of surface EKE from SODA reanalysis (1980–2021) and altimetry (1993–2019) in the



AL region and from MMP (2003–2018) in the BG region are compared together (Fig. 8). In the AL region, EKE was relatively weak showing a slowly increasing over the years before 2003, so we do not discuss it emphatically here. EKE started to increase rapidly since 2005 and peaked in 2009 which lagged behind 1–2 years versus the variety of halocline and it indicated a decreasing until 2012 (AL). Although the EKE from reanalysis is the highest estimate among them, but its fluctuation coincides with the results from altimetry. In the BG region, EKE started to increase rapidly about since 2007 and also peaked in 2009. and it indicated a decreasing until 2014 which was a little different from that in the AL. As a whole, EKE over the years between 2009 and 2014 was both relatively weak in the two regions corresponding with the plateauing of halocline variables. These characteristic shifts of eddy and stratification evolution are both relative to the varying physics of the gyre in the upper layer that indicate a strengthening during the years before 2007 and a possible stabilization since 2008 (Zhang et al., 2016). Despite of some enhancement after experiencing low peaks, recently EKE has not exhibited rapid development on a long timescale but oscillated around a constant level based on current data sets between the central basin and its marginal continental slopes.

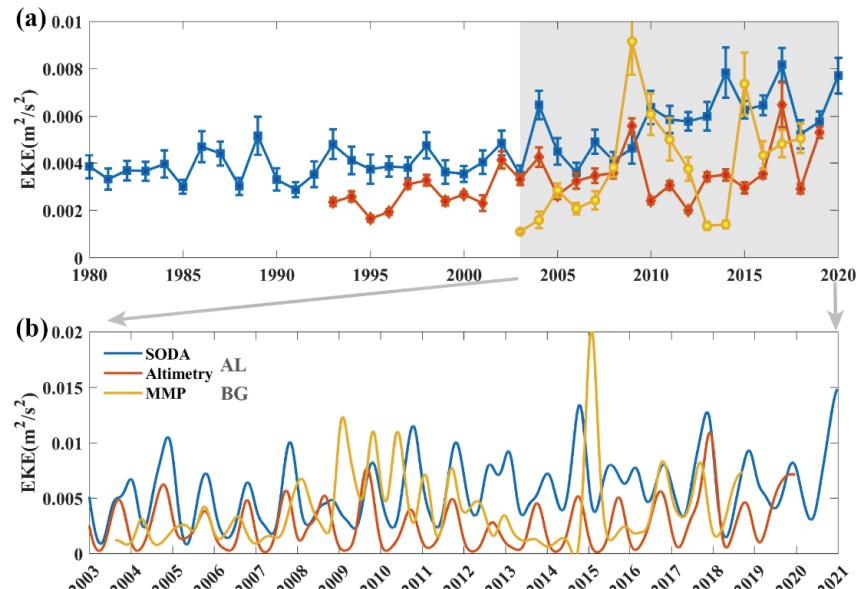

**Figure 8. (a) Annual mean eddy kinetic energy (EKE) from MMP (2003–2018) averaged over 250 m in BG region, altimetry (1993–2019) and SODA (1980–2020) at surface in AL region. Error bars represent 1/10 standard deviation in every year. (b) Time series during 2003–2020 from partial results of (a), that are all smoothed by applying a 100–day low‑pass filter.**

## 5 Eddy modulation in the asymmetric halocline

In the context of gyre variability and the most prominent sea ice losses in the BG region (Timmermans and Toole, 2023), extra wind energy input leads to more active eddies. The halocline as a measure of gyre stability necessarily exhibits significant changes when eddies generate and transport with flows under this background. However, in section 3 the variability of halocline in the BG region demonstrates an apparent reducing of meridional asymmetry. How eddies as a key physical process





modulated the halocline in this phenomenon? In this section we will combine the variety of eddies and EKE analyzed in the section 4 with the varying asymmetry of halocline to shed light on how eddy field modulates in the halocline.

## 5.1 Relationship between geostrophic currents and EKE

The APE and geostrophic currents are both diagnostic variables of the halocline depth (Armitage et al., 2020). Eddies are
generated through dissipating APE and they gradually weaken the slope of isopycnals as well as geostrophic currents. Furthermore, the seasonality of eddy and geostrophic current fields is similar in the Arctic surrounding seas (Armitage et al., 2017). EKE at the southwestern partner of the basin where is the confluence of reversed zonal geostrophic currents is the strongest (Fig 9). And the area with stronger (weaker) zonal currents is companying with relatively weaker (stronger) EKE in the northern (southern) part of Beaufort Sea slope (BSS) region. Along the Alaska coast EKE is higher about 1 order of
magnitude than MKE at most, while in the offshore deep basin MKE is even higher 1 order of magnitude than EKE that is agree with most areas in the Arctic Ocean (von Appen et al., 2022). Recently the domain with strong EKE has been developing gradually departed from coasts, such that EKE in partial areas specially central basin has exceeded MKE (not shown here).

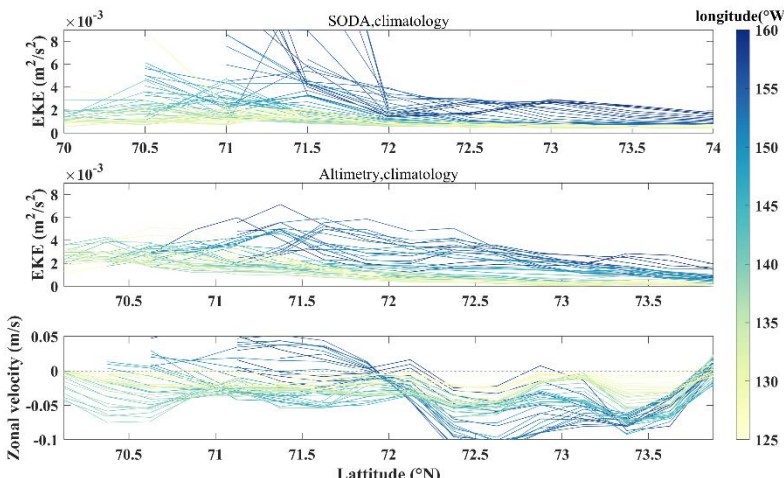

**Figure 9. Climatology meridional eddy kinetic energy (EKE) from (upper panel) SODA reanalysis (1980–2020) and (middle panel)**
**altimetry (1993–2019) in the Beaufort Sea slope (BSS) region. (lower panel) As the same with (a) and (b), but for climatology zonal geostrophic velocity.**

We compare the probability analysis results of EKE and geostrophic velocities averaged in the AL region (Fig. 1b) based on the satellite altimetry for three periods corresponding to the halocline change. From Fig. 10, annual mean EKE was significantly intensified by 17% (26%) from period 1 to period 2 (from period 2 to period 3), simultaneously its main values
within the extent with a probability of 68.4% were also enhanced. Although velocities were both increased in the last two terms, the magnitudes of their increasing are only 15% and 7% much smaller than that of EKE. When EKE in this region remained a sharp increasing in the past the velocity field was increasing more slowly. The rate of velocity change has begun to decrease in recent years while EKE was still increasing rapidly, implying the difference between them was magnified in

recent years. For further information, we explore the relationship between these two variables. Besides that, we find that these

variabilities indicate a strong correlation over the area of we interest (Fig. 10g). The correlation coefficients between EKE and

local geostrophic velocities are mainly negative near the Alaska coast and partial central basin which is verified by their

variation in the AL region while the major correlation coefficients passing a test of the significance level 95% remain highly

positive between 1000 m and 3000 m isobaths along the southwestern margins of the basin, which is likely to be caused by

the continuously enhanced EKE offshore even emerging in the deep basin.

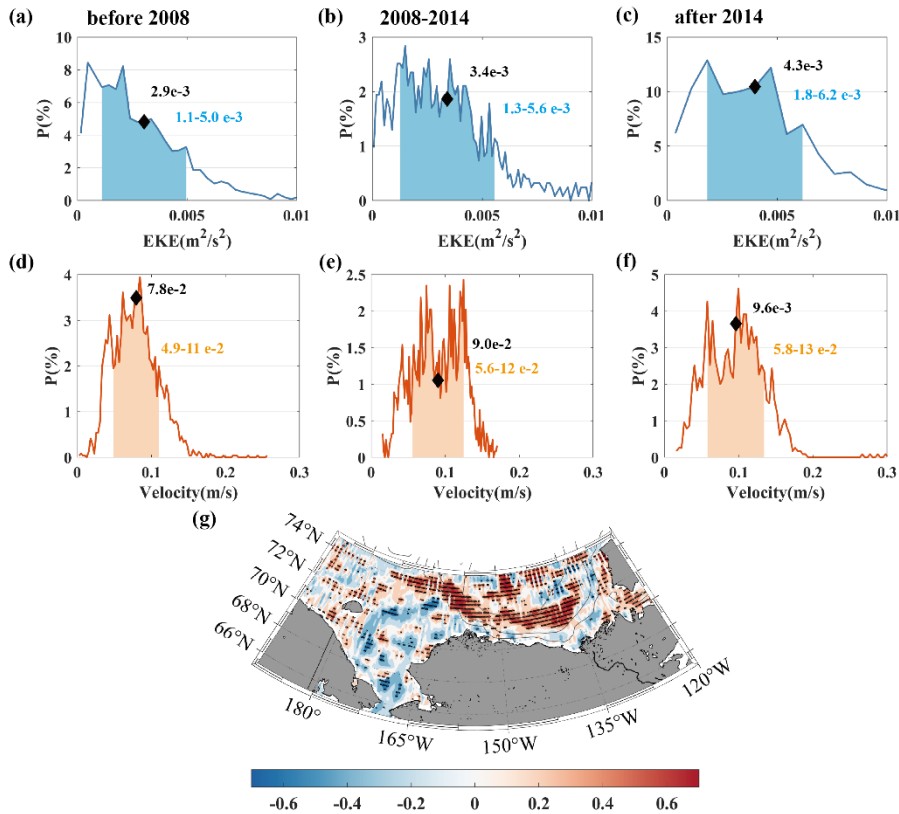


**Figure 10. Probability of (a–c) eddy kinetic energy (EKE) and (d–e) geostrophic velocity in the Alaskan coast (AL) region during (a, d) 1993–2007, (b, e) 2008–2014, and (c, f) 2015–2019. Black diamonds represent mean values in three periods. The range of shading means the extent with a probability of 68.4%. (g) A map of correlation between annual mean eddy kinetic energy (EKE) and local geostrophic velocities in 1993–2019. Black dots indicate all positions that passed a significant test (confidence level 95%).**


## 5.2 Eddy lateral flux: a critical role in modulating the halocline

During 2009–2011 EKE remained at a relatively strong level compared with mean value over the whole period. At the same time, the meridional asymmetry of the halocline geometry has been reduced, and the increasing rate of geostrophic currents has been slowed down. It is known currently that eddies can not only dissipate APE but also hinder the accumulation of



freshwater. As is discussed in section 3, the halocline vertical structure tends to be meridionally symmetrical in the BG region
        in recent years, which is proved by in situ observation from MMP and CTD observations. Here we also find that this varying
        structure can be well replicated schematically through SODA reanalysis (Fig. 11a), although the results from SODA
        overestimate the depth of the halocline to a certain extent with an error of 30–40 m near the central basin. The changes of the
        halocline structure and depth at each side in the three periods obtained from SODA showed a strong consistency with the
results from CTD, which is verified that northern halocline was upper than southern part before 2008 and then the halocline at
        each side along the meridional transect remained a similar level after 2014.

        Aiming to explore what a critical role did eddies paly in the halocline, we analyze the eddy streamfunction evaluated by Eq.
        (5) over a long-term scale based on SODA. As is shown (Fig. 11b), in the first period when the Eady timescale was relatively
        large over the long term, the abnormal salinity in the mixed layer and the halocline was both positive. Combined with the
distribution pattern of eddy streamfunction, the eddy thickness flux was positive at surface due to the southward propagation
        of low-salinity water, and above the base of the halocline was mainly distributed at the edge of the gyre. Low-salinity water at
        subsurface in the south near the continental slope spread northward but in the north close to the deep basin it spread southward,
        which formed a central-converging pattern, so finally resulting in southern halocline much lower than the north at that time.
        In the second period when a transformation appeared in the upper layer, the Eady timescale was decreasing meaning the
enhanced baroclinic instability in the BG. The mixing layer showed a low salinity anomaly and the pattern of eddy thickness
        flux indicated a northward propagation of low-salinity water. Meanwhile, there was an overall deepening of halocline depth.
        In the third period, significant low salinity anomaly in the halocline has been transferred from surface to subsurface. Besides,
        the main spatial pattern of eddy flux in this period was extremely similar to that in the former period with obviously
        strengthened. The low brine transmission caused by eddies replenished the surface freshwater in the north. Above the halocline,
in the main range of 71°–79° N surrounding the central gyre, the convergence of anormal low salinity was extremely strong.
        As a whole, the freshwater redistribution induced by these transportations due to eddy lateral flux are contributing to
        significantly diminished the meridional asymmetry of the halocline. Some of the low-salinity water continues to spread
        northward, which is coinciding with the northward expansion of the gyre mentioned in a recent study (Bertosio et al., 2022).



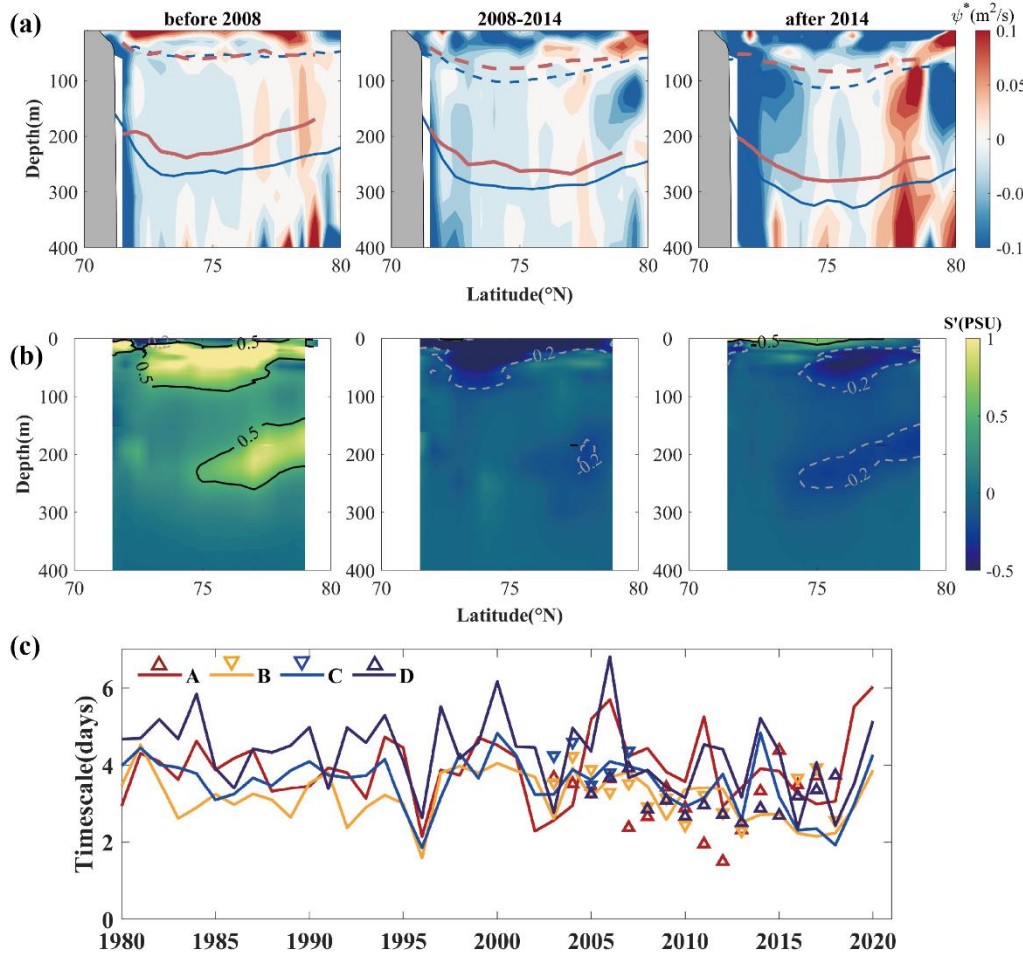

**Figure 11. Transects of (a) eddy streamfunction (unit: m²/s) and (b) abnormal salinity (unit: PSU) relative to the whole term averaged in 2004–2007 (before 2008), 2008–2014 and 2015–2020/2021 (after 2014), that are calculated from SODA simulated now till 2020 and CTD observed till 2021, respectively. The dashed (solid) lines indicate the processed annual mean depth of isopycnal surface σ = 25 (27.4) kg·m⁻³ representing the top (base) of halocline from SODA (bule lines, selected data are from September to October which is mostly consistent with the observed data for CTD deployment) and CTD (red lines) during three periods. (c) Annual mean time series of Eady timescale calculated from SODA (solid line) and MMP (triangle) at four moorings.**

## 6 Summary and discussion

The main objective of this research is to explore how long-term variations of eddy activity influence the spatial-temporal variability of halocline under the BG system. In this study, our analyses of halocline based on in situ hydrologic data including MMP from moored observations and CTD under the BGEP project both showed that northern and southern depth of isopycnals have deepened in different degrees in the last nearly two decades. The halocline depth and strength are both significantly increased among the deep basin and continental slopes (Kenigson et al., 2021; Zhong et al., 2019). The halocline in the south near 74° N has been deepened by ~40 m while that in the north near 77° N has been deepened by ~70 m over the years 2003–





2018. After 2014, the difference of halocline depth at the two sides of the basin was nearly negligible. The meridional asymmetry of halocline with halocline depth lifting to the north initially was shifted to a final nearly symmetric structure.

Furthermore, we investigated the spatio-temporal variability of eddies and EKE between the central gyre and continental slope to try to clarify why the halocline changed asymmetrically. There were 37, 40, 7 and 43 eddies detected in the upper layer at mooring A–D, 98% of which were anticyclones. EKE at the southwestern corner was much stronger after 2008 than that in the previous period but it remained relatively stable latter, which was consistent with Eady timescale. EKE above the halocline is intensified antecedently in the south compared with that in the north from mooring measurements, which is demonstrated

by the long-term varying EKE distribution in the southwestern Arctic relating to the direction of eddy propagation from where they are generated. With halocline depth varying, the number/strength of eddies at different sites as well as EKE at key regions are exhibiting considerable interannual fluctuations that are also related to the movements large-scale circulation. When EKE is enhanced along the Chukchi/Beaufort continental slope because of baroclinic instability, it is gradually developed toward central basin, which is agree with its intensification in the interior gyre from observations. Under the modulation of

continuously increasing EKE provided by multiple data sets in the key region before 2009, the halocline depth experienced a deepening and then a lifting or a stagnate phase in the BG region, and the increasing of geostrophic flows also slowed down. It is worth noting that high EKE region is close to the reversal currents with relatively weak flows there.

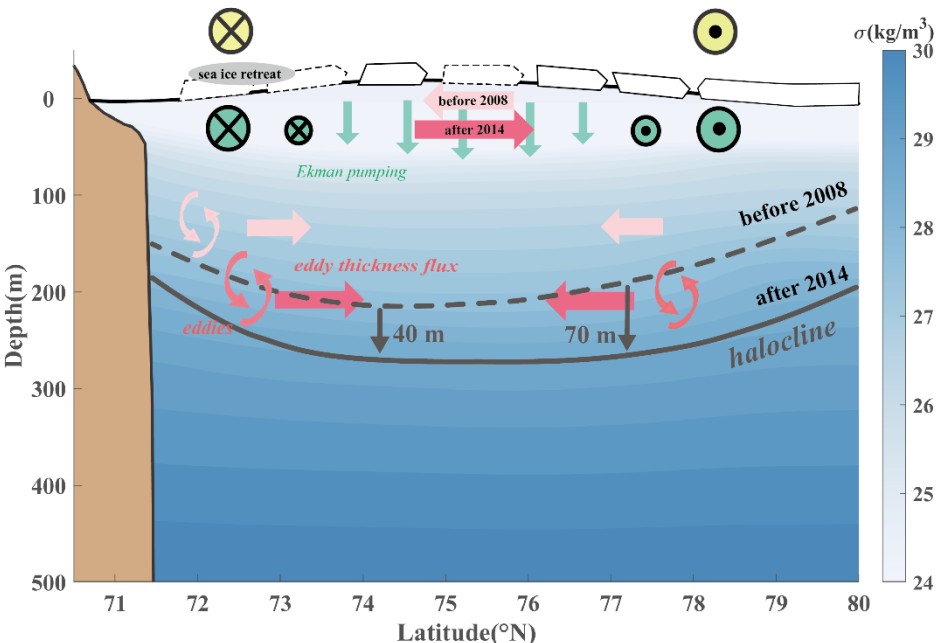

**Figure 12. Schematic diagram under the BG system referred to the transect of 150°W indicating the recent eddy modulation in the**
**halocline. Shading is the climatology potential density from 2005–2017 WOA. Light (Dark) green arrows represent the eddy thickness momentum before 2008 (after 2014).**

Overall, the credible results revealed that the eddy flux, playing a critical role in modulating the halocline, has adjusted the vertical structure of halocline through affecting the freshwater redistribution in the past years comparing the initial period with

the latest (Fig. 12). At the moment, meridional asymmetry of halocline of the BG is distinctly diminished attributing to
modulation of eddy lateral flux. At first time, the eddy flux was mostly positive above the mixed layer meaning the southward
propagation of low-salinity water, which explain the tilted structure of halocline. In the latest period, the eddy flux above the
mixed layer was remarkably negative meaning the northward propagation of freshwater and it formed an extremely strong
convergent center in the halocline. A series of processes promoted the surface low-salinity water transmit to the northern basin
and it can be beneficial for the confluence of freshwater in the halocline at depth from two sides, which adjusted the meridional
distribution of halocline from asymmetry to relative symmetry.

To date, previous researches hypothesized that the accumulation of freshwater driven by Ekman pumping is balanced by the
rectified effect of mesoscale eddies for stabilizing the circulation (e.g., Davis et al., 2014; Manucharyan and Spall, 2016), not
yet probing too much the eddy dynamics for modifying halocline asymmetry. This paper provides a perspective for
understanding the long-term changes of stratification structure and eddy field in the BG and the relationship between them.
We expect it can improve the knowledge of large-scale circulation and mesoscale process under the background of rapid
changes in the Arctic. It is still necessary for us to apply for high resolution simulation and observations across the gyre to
obtain a comprehensive understanding of interior variation among different physical processes applying to promote scientific
development in the BG dynamics.

**Data availability**

The gridded satellite altimeter data (product identifier: SEALEVEL_GLO_PHY_L4_REP_OBSERVATIONS_088_047) is
freely        made        available        by        the        Copernicus        Marine        Environmental        Monitoring        Service
(https://data.marine.copernicus.eu/products). Observations including CTD and MMP profiles are collected and made available
by        the        Beaufort        Gyre        Exploration        Project        based        at        the        Woods        Hole        Oceanographic        Institution
(https://www2.whoi.edu/site/beaufortgyre) in collaboration with researchers from Fisheries and Oceans Canada at the Institute
of Ocean Sciences. The SODA data set is from the Ocean Climate Lab at the University of Maryland
(https://www2.atmos.umd.edu/~ocean/index.htm).

**Author contribution**

DL provided the initial scientific idea and financial supports. DL and JL together conceived the idea for the present study. JL
and ST collected all available data sets. JL processed the data, plotted the results and wrote the first version of the manuscript.
All authors reviewed and edited the manuscript to its final version.



**Competing interests**

The authors declare that they have no conflict of interest.

**Acknowledgements**

The authors acknowledge Service Woods Hole Oceanographic Institution for making mooring data and in situ observations
available. We thank the Copernicus Marine Environmental Monitoring for providing altimetry for providing altimetry data.
We appreciate the Ocean Climate Lab at the University of Maryland for providing data assimilation.

**Financial support**

This study is supported by the National Natural Science Foundation of China (NSFC, grant no. 41576020, 42076228 and
42230405).

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
