# Peer review of "Long-term eddy modulation inhibited the meridional asymmetry of halocline in the Beaufort Gyre"

_EGUsphere, 2023_

## Author Comment (AC2)

**Response to reviewer for manuscript "Long-term eddy modulation inhibited the meridional asymmetry of halocline in the Beaufort Gyre" by Lu et al.**

The author comment is presented in the following sequence: (1) itemized comments from the reviewer in black, (2) author's response in blue, (3) quotations from the revised paper in indented *blue italic*.

Anonymous reviewer #1:

This paper explores the relationship between the varying eddy field and the changes in the shape and depth of the Beaufort Gyre halocline in the past two decades. The authors used in-situ observations, altimetry, and reanalysis data, to describe the changes in the halocline structure, with an emphasis on the meridional asymmetry. They then examined eddy activities, from both individual eddies and kinetic energy perspectives, and connect them to halocline structure via available potential energy. The study found that eddies played an important role in redistributing fresh water and thus adjusted the halocline structure through analysis of eddy fluxes. Despite the paper's intriguing ideas, the writing is unclear, making it very difficult to follow. The overall flow needs improvements. Therefore, I cannot recommend this paper to be published in its current form.

Thanks to the reviewer for recognising the value of our work and constructive comments. We genuinely appreciate the time and effort you have dedicated to thoroughly reviewing our work. We agree the thorough suggestions to improve our manuscript. The overall flow has been carefully polished to clarify our scientific results. We have also made more detailed explanations in our responses and revised our manuscript.

Minor points:

Line 132: Could the authors explain how eq (3) relates to this sentence?

Thanks for pointing it out. The author apologizes for the wrong text provided. The Eq. (3) is corrected to Eq. (2).

Line 146: This is not the right place to insert citation.

Thanks for pointing it out. We corrected it.

*(Line 149-154)*

*...... $\psi^*$ is represented as*

$$\psi^* = \frac{\overline{V'S'}}{S_z} = -\frac{\overline{w'S'}}{S_y} \qquad (5)$$

*where $\overline{V'S'}$ is the average meridional eddy salt flux and $\overline{S_Z}$ is the average vertical salt gradient (Manucharyan et al., 2016; Manucharyan and Spall, 2016; Manucharyan and Isachsen, 2019; Marshall and Radko, 2003).*

Line 205-206: Mooring C ends before 2008. How can we make a conclusion about the shape in the northeast and northwest in recent years?

Sure, we are also aware of the limit of mooring observations but the results still show that the time series of halocline depth and thickness from MMP in the northwest (mooring B) and northeast (mooring C) are well overlapped before 2008. We also supplement the results from CTD (dot), which is better for better understanding the difference between northern and southern sites over the whole period. The evolutions of halocline at the southwestern (mooring A) and southeastern (mooring D) sites are similar. Likewise, The evolutions of halocline at northern sites are also similar. Compared with MMP results, the mean relative errors of CTD on halocline depth (thickness) are 2.0% (3.4%), 4.4% (7.0%), and 1.0% (3.0%) for moorings A, B, and D, respectively. Figure 2 is modified shown below.

[Figure]

Figure 3: APE at a single point is not meaningful. It represents the slope of the halocline in a region, not a single point. The reference density in the APE equation is related to some mean density across the field. Authors should make it clear.

Thanks for the advice. The Eq. (3) is corrected to estimate the APE in the BG area. Figure 2 in our revised paper is also modified that is shown in the previous item.

*(Line 147-150)*

*The calculation of APE here is following Eq. (3):*

$$APE = PE - PE_{ref} = \iiint_{z_{ref}}^{surface} g[\rho(z) - 1027.4]zdz \, dA \qquad (3)$$

*where $z_{ref}$ represent the depth of the halocline lower boundary, A is the gyre area (Armitage et al. 2020; Polyakov et al., 2018; Bertosio et al., 2022).*

Figure 6: The lengths of data available are determined the days of open ocean. Will different data lengths impact the results?

The altimetry observations are available in summer and fall when there is nearly no sea ice covered in the area we interest. EKE is strongest in summer/fall and weakest in winter (Wang et al., 2020; Manucharyan and Thompson, 2022). Using altimetry, we can well reap the long-term variability of major EKE.

Figure 6: The plot is from satellite data, so I assume it indicates surface EKE. How is it related to the previous paragraphs in which subsurface EKE are discussed?

We are trying to discuss EKE at the surface and subsurface together to better clarify the EKE variability in the upper layer. The order of figures was changed in this section for better understanding. Original Figure 6 was changed to Figure 7. Figure 7 shows the evolution of EKE pattern. After 2008, EKE was strengthened along southwestern slope than before, agreeing with more active halocline eddy activity in the BG region from moored observations.

Line 306: This statement made based on observations from two points. Other factors could be responsible. For example, eddies were generated near the mooring site; or the eddies simply did not pass by the mooring.

Thanks for pointing it out. This statement was modified. There are main two factors emphasised, including the eddy distribution and eddy transportation.

Figure 10: How is the probability estimated?

The estimation of probability was added. And related expressions were added as follow:

*(Line 381)*

*We compare the probability analysis results of EKE and geostrophic velocities averaged in the AL region based on the satellite altimetry in three periods (Fig. 10), which is estimated by statistical frequency of the area mean time series in every period.*

Line 399: Eady timescale is not interpreted previously.

Meaning and calculation of Eady timescale are interpreted in section 2.2. It estimates the oceanic baroclinic instability.

Major point:

The authors described both surface and subsurface EKE. There could be asymmetry in the surface sea level and subsurface halocline. How does the surface EKE relate to the subsurface halocline shape?

Thanks for suggestions. We try to clarify the relationship between eddy activity and halocline changes which are related to Beaufort gyre variation, and we have revised the explanation in sections 4 and 5.

Based on the previous works, the surface eddy activities responding to the extra wind energy input are linked to gyre stability (Armitage et al, 2020), meanwhile the subsurface EKE from baroclinic instability and APE release are also linked to gyre stability (Manucharyan and Spall, 2016; Manucharyan et al., 2016). Since the two processes both modulate halocline variability in the BG, eddy activities at the surface and subsurface are needed to discuss together to explain their impact on oceanic stratification in the upper layer. The surface EKE can hinder the development of mean kinetic energy (MKE) and slow down the increasing rate of mean currents, which promoted the BG stabilisation (section 5.1). From this perspective, eddy activity can weaken the gradient of surface sea level to inhibit surface geostrophic currents, but it is not the main point of the study. What we are concerned about is that surface and subsurface eddy activities jointly influenced oceanic stratification by inhibiting surface mean flows and promoting APE release above the halocline layer related to BG stabilisation. When mean kinetic energy and potential energy are more stable in the final period (after 2014) than before, eddy modulation leads to freshwater redistribution in the upper layer through enhanced eddy lateral fluxes, which inhibited the meridional asymmetry of halocline changes (section 5.2). It is shown that freshwater distribution has an impact on the variability of halocline thickness and strength (Bertosio et al., 2022). Therefore, we concluded that eddy fluxes, generating low-salinity water transportations in the upper layer (Fig. 11b), resulted in a smaller difference in halocline depth between north and south of BG.

To show how the writing could be improved, I will provide some wording issues in the first 50 lines. Please note that this list is not exhaustive, but rather serves as a guide for improving the paper's overall clarity.

Thanks for the guideline. We checked the whole manuscript again for better clarification.

Line 10: Please specify the "varying eddies". Number, shape, velocity?

Thanks for pointing it out, we have specified.

Line 16: This long sentence is confusing. Maybe it would be clearer if breaking into several shorter sentence.

Thanks, this long sentence was divided into several segments.

*(Line 16)*

*In the meantime, eddy activities in the upper layer from the southern margin of BG to the abyssal plain have been enhanced. Moreover, eddy-induced low-salinity water transportations have been continuously increasing towards the central basin as halocline structures on either side of the basin reach a nearly identical and stable regime.*

Line 16: The second "Eddy" should not start with capital letter.

Thanks for pointing it out, it was corrected.

Line 30: This sentence is not grammatically correct.

Thanks for pointing it out. This sentence was reorganised.

*(Line 29)*

*Meanwhile, increased active ocean–atmosphere interactions and mesoscale processes in the Canada Basin (CB) due to the emergence of broader open areas have attracted increasing attention.*

Line 37: It should be "paid attention" instead of "payed attention"

Thanks for pointing it out, it was corrected.

Line 39: "The increase of isopycnal slope with depth" is a new piece of information and it should be introduced before directly describing the mechanism.

Thanks for this recommendation. "The increase of isopycnal slope with depth" explained the interior increased PWW thickness. This paragraph was reorganised.

*(Line 43-47)*

*Pacific Winter Water (PWW), which lies above the eastern Arctic origin lower halocline water, is recognized as a component of the western Arctic halocline (Shimada et al., 2005). Observations indicated that PWW layer generally deepened during 2004–2018 while isopycnal layer thickness increased (Kenigson et al., 2021). Likewise, there was an isopycnal deepening by 70 m during 2004–2011 (Zhong et al., 2018), suggesting a spin-up of the gyre. The isopycnals are increasing with depth, which can be attributed to the eddy-induced stream function, explaining the interior increased PWW thickness (Kenigson et al., 2021).*

Line 42: "Likewise" used here is confusing. The previous and this sentence do not have characteristics in common.

Thanks for pointing it out, it was corrected. These sentences are modified for more coherence.

Line 43: Should be "identified as".

Thanks for pointing it out, it was corrected.

Line 46: high resolution eddy-resolving?

Thanks for pointing it out, it was corrected.

Line 47: "etc" should not be used.

Thanks for pointing it out, it was corrected.

Line 48: "focused on" instead of "focus".

Thanks for pointing it out, it was corrected.

Line 48: "Moreover" is not correctly used here. The following sentence changes the topic.

Thanks for pointing it out. The following sentence is the same topic on the vertical distribution of eddy activity. This sentence was rephrased for more coherence.

Line 50: "even they may extend"?

This sentence has been rephrased.
> *(Line 52)*
> *Eddies are mainly concentrated in the subsurface (30-300 m) even though they can extend to thousands of metres in depth (Zhao et al., 2014; Zhao and Timmermans, 2015) ⋯⋯*

Reference:
Armitage, T., Manucharyan, G. E., Petty, A. A., Kwok, R., and Thompson, A. F.: Enhanced eddy activity in the Beaufort Gyre in response to sea ice loss. Nat. Commun., 11, 1-8. https://doi.org/10.1038/s41467-020-14449-z, 2020.
Bertosio, C., Provost, C., Athanase, M., Sennéchael, N., Garric, G., Lellouche, J. M., Bricaud, C., Kim, J. H., Cho, K. H., and Park, T.: Changes in freshwater distribution and pathways in the Arctic Ocean since 2007 in the mercator ocean global operational system. J. Geophys. Res. Oceans, 127, e2021JC017701. https://doi.org/10.1029/2021JC017701, 2022.
Manucharyan, G. E., and Spall, M. A.: Wind‐driven freshwater buildup and release in the

Beaufort Gyre constrained by mesoscale eddies. Geophys. Res. Lett., 43, 273-282. https://doi.org/10.1002/2015GL065957, 2016.Manucharyan, G. E., Spall, M. A., and Thompson, A. F.: A theory of the wind-driven Beaufort Gyre variability. J. Phys. Oceanogr., 46, 3263-3278. https://doi.org/10.1175/JPO-D-16-0091.1, 2016.

Manucharyan, G. E., and Thompson, A. F.: Heavy footprints of upper-ocean eddies on weakened Arctic sea ice in marginal ice zones. Nat. Commun., 13, 1-10. https://doi.org/10.1038/s41467-022-29663-0, 2022.

Wang, Q., Koldunov, N. V., Danilov, S., Sidorenko, D., Wekerle, C., Scholz, P., Bashmachnikov, I. L., and Jung, T.: Eddy kinetic energy in the Arctic Ocean from a global simulation with a 1‑km Arctic. Geophys. Res. Lett., 47, e2020GL088550. https://doi.org/10.1029/2020GL088550, 2020.

---

## Author Comment (AC3)

**Response to reviewer for manuscript "Long-term eddy modulation inhibited the meridional asymmetry of halocline in the Beaufort Gyre" by Lu et al.**

The author comment is presented in the following sequence: (1) itemized comments from the reviewer in black, (2) author's response in blue, (3) quotations from the revised paper in indented *blue italic*.

Anonymous reviewer #2:

Review of "Long-term eddy modulation inhibited the meridional asymmetry of halocline in the Beaufort Gyre" by Lu et al.

This paper uses a combination of satellite, in-situ and reanalysis data to investigate changes to the structure of the halocline in the Canada Basin over time. The paper begins by looking at changes in halocline depth, halocline thickness, and available potential energy from CTDs and moorings in the region to determine three periods of distinct behaviour. It then analyses both EKE and individual eddy properties at the four moorings and how they have varied over the three periods, before zooming in to a small region that experiences significant changes to EKE over the time period. The paper ends with an analysis of an eddy streamfunction for each period and relates it to salinity changes to explain the differences in the halocline between each period.

The paper has a lot of information contained within it, but I found it very hard to follow in many places. Some of the figures had long descriptions of details without reference to the relevant subpanel or feature being described, and the main take-home message of what the reader is meant to gain from the figure is often missing. This makes it difficult to understand the context. There are also a number of typos and grammatical errors which need to be corrected – I have not addressed them in the comments below but the manuscript should be checked thoroughly and rephrased in a number of places. I appreciated the instances where the authors stated the question they were addressing in the upcoming subsection, and feel it would be beneficial to do this much more often in the text to help the reader. I do not think the paper is publishable in its current form, but do think there are some interesting ideas that could be of interest to the community if they were presented in a more coherent way.

Thanks to the reviewer for your valuable feedback and suggestions. We sincerely appreciate the time and effort you have put into reviewing our work. Your comments have been incredibly helpful in improving the quality of our research. We are grateful for your illuminating insights and constructive criticism you provided, which has allowed us to address certain weaknesses and refine our findings. The overall flow has been carefully polished to clarify our scientific results. We have also made more explanations in our responses and revised our manuscript.

I have some points to consider based on the paper in its current form, which I feel should be addressed before the paper is resubmitted.

- The paper aims to understand why the BG has stabilised in recent years. But the asymmetry that is described is based on a) moorings in the south versus moorings in the north of the basin (which are located at around 74-75N and 78N respectively, based on Figure 1b), and b) CTD stations which head north along a transect to 79N. The paper refers to Bertosio et al (2022) and Regan et al (2019) when describing gyre asymmetry and a northward expansion, and also notes a shift in the BG found by Moore et al. (2018). Given this acknowledgement that the gyre is not always in the same place or has the same size/strength, it is surprising that there is no discussion of how choosing a static section or static moorings that are limited in their northward extent could affect the perceived loss of asymmetry of the gyre. The gyre has deepened in the portion of the basin captured by the observations, but the deepest part of the section has also moved further north – so the sloping isopycnals are likely now further north than the section shows. It is not known from this data whether they are steeper or flatter than before, but I feel this should be acknowledged in the paper (does the SODA data show this if you look further north?)

We thank the reviewer's suggestions. We agreed that BG area and size, depending on dynamic ocean topography, significantly changed with seasonal and long-term variabilities (Regan et al., 2019). In our study, the main point is the long-term variability of the halocline relating to the main BG area. Thus, we need to determine a referential BG region. The observations are limited in the area we interest, resulting in directly tracking the gyre area for analysis considerably challenging. The fundamental BG area is located in the Canada Basin even if the whole size is changing for each year. Therefore, we choose the static section and moorings in our study to reap the long-term variability in the fundamental BG area.

We replaced the WOA18 in the previous version with the newest WOA23 including 1990-2020 climatology hydrography, for discerning the fundamental BG area to analyse halocline variability. We supplemented a map illustrating the climatology halocline depth (Fig. 1a). Additionally, a BG box (referred to as the pink box), along with the climatology BG center based on Regan et al. (2020), has been marked. This BG box is defined as the region between 70.5-80.5°N and 170-130°W, bounded by the 300 m bathymetry. The centre of the mean gyre from 1990 to 2014 is situated at 74.74°N and 150.62°W. Despite the variations in BG area over the past years, the chosen BG box effectively captures the primary pattern of halocline characteristics that relates to core BG region, encompassing the deepest region in the western Arctic. The definition of fundamental BG area is widely adopted by many studies on BG (e.g., Doddridge et al. 2019; Manucharyan and Spall, 2016; Regan et al., 2019; Timmermans and Toole, 2023).

The placement of the four moorings at the corners of the BG box allows for a better understanding of the overall BG halocline variability, which is used by former research for analysing BG stratification (e.g., Kenigson et al., 2021; Manucharyan and Stewart, 2022; Zhong et al. 2019). We specifically selected the static section along 150°W for further analysis. To supplement the discussion, we also analysed the section along 140°W and made a comparison (**Fig. 4, shown below**), as both sections traverse the deepest part of the BG halocline. Timmermans and Toole (2023) found that theses two sections through BG have similar hydrographic structures. Our results also demonstrate similar shifts along these two

sections. However, we think the section along 150°W offers a more representative perspective since the BG centre is positioned between 150°W and 140°W for most years, with closer proximity to 150°W (Regan et al., 2019). the change is more significant and the halocline layer is much thicker along 150ºW than along 140ºW because the gyre centre is closer to the 150ºW longitude. Thus, we choose the 150°W transect for better discerning variability of the halocline layer.

[Figure]

We discussed halocline slope in the BG through supplementary APE evolution in the BG box that can represent the slope of the halocline in a region. The halocline slope in BG region was increasing before 2010. However, in the final term, halocline was flatter than before with APE decreasing after 2010 and remained at a relatively stable level after 2014, which represented the flattening of halocline .

- The discussion around EKE and individual eddies is unclear. In particular, surface and subsurface (from 50m down) EKE seem to be shown on the same plot, and surface-generated EKE from wind input is linked to gyre stability (e.g. lines 347-349) even though the eddies associated with baroclinic instability are generated in the halocline. Care should be taken when associating these.

Thanks for the reviewer's suggestion. In the revised manuscript, the discussion in section 4 is checked carefully to better clarify our main points.

We are talking about the spatiotemporal variability of eddy activity using limited observations. On the one hand, from moored observations we focus on the variability of individual eddies at the subsurface due to a lack of surface observations. The number and kinetic energy of individual eddies here are used for estimating eddy activeness and number. On the other hand, we put more attention to analyse EKE variability from multiple datasets to comprehend thoroughly eddy strength evolution in the upper layer for insufficient moored observations.

The surface eddy activity from extra wind energy input is linked to gyre stability (Armitage et al, 2020), and subsurface EKE from baroclinic instability and APE release is also linked to gyre stability (Manucharyan and Spall, 2016; Manucharyan et al., 2016). We think the long-term cumulative effects from transient eddies can influence the mean states of halocline structure. And it is necessary to explore the spatiotemporal variability in the eddy field before discussing its effects on the halocline structure. Thus, we put them on the same plot to discuss their similar variability at the surface and subsurface together. The surface and subsurface EKE peaked in 2009 and experienced a low ebb in 2010-2014. With APE in the BG decreasing continuously over the years 2010-2014, after 2015 EKE increased again and remained at a stronger level than before. In section 5, we discuss eddy modulation detailedly. The surface EKE can hinder the development of mean kinetic energy and slow down the increasing rate of currents, which promoted the BG stabilisation (section 5.1). What's more, surface and subsurface eddy activities jointly influenced the freshwater redistribution through eddy lateral flux (section 5.2), which inhibited the meridional asymmetry of halocline.

- In terms of structure, the flow is broken after Figure 5 (Mooring-based EKE and eddy counts) to Figure 6 (maps of EKE from other datasets), then back to eddies from moorings in Figure 7, then back to the maps to identify a key region to zoom in on for Figure 8. Is there a reason that we jump between datasets and region size? It might flow better if all of the information from moorings was put together. I found the jumps from Figure 6 to 7, then 7 to 8, quite confusing, so maybe that would help to make it clearer

Thanks for this suggestion. We reorganised section 4 and modified the order of figures based on the suggestion. In section 4.1, we just discuss mooring-based eddy counts and EKE profiles. In section 4.2, we analyse EKE variability from multiple datasets, especially on the long-term variability, including surface EKE from altimetry and SODA, and subsurface EKE (averaged over 250 m) from MMP. The order of figures in section 4 is changed accordingly.

- Introduction: there is a lot of information that has not been fully synthesised. In particular, the second paragraph is very long and detailed with a lot of different threads. I would suggest splitting into multiple paragraphs, perhaps one describing the vertical structure and one based on eddies. In general, there is a large amount of information on eddies in the introduction paragraphs – perhaps it can be streamlined, or reordered to group similar themes together.

Thanks for this suggestion. This paragraph has been synthesised in the revised version. This paragraph was divided into three parts, including vertical distribution, horizontal distribution and long-term evolution of eddy number and EKE.

Line 42: The Pacific Winter Water layer is mentioned without describing how it fits into the vertical structure. It should be introduced first

Thanks for pointing it out, we supplement the description of Pacific Winter Water layer.

*(Line 42)*

*Pacific Winter Water (PWW), which lies above the eastern Arctic origin lower halocline water, is recognised as a component of the western Arctic halocline (Shimada et al., 2005).*

Line 50: What depth range is meant by "subsurface"?

Eddies are found concentrated in the halocline (Zhao et al., 2014; Zhao and Timmermans, 2015). "Subsurface" means the halocline depth range about 30-300 m.

*(Line 53)*

*Eddies are mainly concentrated in the subsurface (30-300 m) even though they can extend to thousands of metres in depth (Zhao et al., 2014; Zhao and Timmermans, 2015)······*

Line 132: is (3) the correct equation reference here? It hasn't yet been introduced

Thanks for pointing it out. Sorry, it is a clerical error. The author apologizes for the wrong text provided. The Eq. (3) is corrected to Eq. (2).

Line 155: Simth, 2007 should be Smith, 2007. Eddies are only a part of the EKE which also includes deviations from a mean current. How much EKE is not attributable to eddies? I.e. how much is not due to eddy genesis? That might affect the assumption that it is correlated with baroclinic growth rate. How much EKE do you miss by only having SODA at ½ degree resolution?

Thanks for pointing it out . To discern how much EKE is not due to eddy genesis, we make a comparison between EKE and kinetic energy from eddies ($KE_{eddy}$) in modified Fig. 6 (shown below). Approximately 50% of EKE is due to eddy genesis.

[Figure]

South of the CB is populated with a large number of cold core, anticyclonic halocline eddies. Eddy genesis in area we interest is more correlated with baroclinic instability rather than barotropic instability. Baroclinic conversion term associated with eddy flux is dominant. Integrated barotropic energy conversion over the Beaufort slope sea section is about an order of magnitude less than the integrated baroclinic conversion term (Spall et al., 2008). Hence, in our study, Eady growth rate correlated with baroclinic instability is applicative.

EKE is just the estimation of eddy strength. It is unrealistic for us to provide how much EKE precisely from SODA reanalysis. We compare the climatology EKE in the Beaufort slope sea region from altimetry (¼ degree resolution) and SODA (½ degree resolution). The magnitude of EKE is comparable within the two datasets. EKE range is approximately 4-6×10$^{-3}$ m$^2$/s$^2$ in the southeast of Beaufort slope (Fig. 9). The correlation coefficient of EKE long-term time series between altimetry and SODA during overlapping years is 0.48 (confidence level 95%). Because altimetry observations are just at the surface, three-dimensional SODA is necessary to be used in section 5 for analysing eddy fluxes.

Lines 160-161: Section 3 talks about the asymmetry of the halocline being the focus of the article, but this was only mentioned briefly amongst all of the text about eddies. I understand that EKE is being investigated to explain the asymmetry, but feel the asymmetry needs to be introduced more thoroughly first – why do we care that it's asymmetric or not?

Thanks for this recommendation. The introduction to the asymmetry of the halocline is accentuated in section 4 first.

*(Line 249-254)*

*With BG spin-up and regional sea ice retreat, mesoscale eddies are responding to dissipate extra energy input and influence the energy redistribution (Armitage et al, 2020). It is speculated that the eddy genesis is related to APE accumulation and release in the BG region, which can influence the vertical structure of the internal halocline (Manucharyan and Spall, 2016; Manucharyan et al., 2016). In the final period, the developments of meridional asymmetry in the halocline layer and APE within the BG box have been inhibited. Under this background, the spatiotemporal variability in eddy activity, needed for a comprehensive understanding, is discussed in this section.*

Line 169: I think by "void measurements" you mean "lack of measurements"?

Thanks for pointing it out, we changed it.

Line 174: what do you mean by "30m company"?

Thanks for pointing it out, this sentence was rephrased.

*(Line 184)*

*The thickness of the halocline in the southern part of the basin (moorings A and D) increased by approximately 30 m with the halocline base deepening by approximately 40 m.*

Line 182: Does "in final" mean "in the final period"? Or "finally", as in the final point being made? I am not sure what is meant by "homogeneously distributed", or what differences are being described as reduced compared to what.

Thanks for pointing it out. It means "in the final period". "homogeneously distributed" means values for different sites are at a similar level. This sentence is rephrased.

*(Line 193)*

*The halocline thickness and depth between every site tend to be at a similar level in the final period and those differences are smaller than in the first period.*

Line 191: "improving" is not correct here. "Increasing"?

Thanks, it was corrected.

Lines 194-195: what are partial variables?

"Partial variables" mean "halocline variables". And it was modified.

Table 1, and related text: what is the significance of these trends? Some are very small, and there is clear variability in the time series. For example, lines 177-179 state "A negative trend

of halocline depth is clearly during 2008– 2014 in the southern sites of the basin (moorings A and D)" but in the table Mooring D only deepens by 0.35 m/yr – is it statistically significant? Is the short-lived deepening in early 2009 having an effect on this trend?

These trends are all statistically significant and all pass significance tests. The confidence levels of these trends are all exceeding 99%. The significance is added to our revised manuscript. The negative trend means halocline depth is lifting during that period, which is not comparable with the deepening trend, because the deepening trend is dominant over the whole period. The short-lived deepening is in later 2008 belonging to period 1, which does not have an effect on the trend of period 2.

Lines 205-206: "According to section 3.1, we find the main differences of evolution only between northern and southern basin are obvious, which is not completely identical with previous findings." What specifically is different from previous studies?

Previous observations have revealed that isopycnals have deepened at different rates in the northwestern and southeastern parts of the basin during 2002–2016 (Zhong et al., 2019). Here we find halocline depth in south and north has been deepening at different rates. The meridional difference between north and south is more obvious. As shown in Fig. 2 and Table 1, the evolution of northwestern (mooring A) and northeastern (mooring D) halocline depth time series are similar.

*(Line 212)*

*According to section 3.1, we find that the major differences in evolution only between the north and south of the basin are obvious, which is not completely identical to previous findings. Previous observations have revealed that isopycnals have deepened at different rates in the northwestern and southeastern parts of the basin during 2002– 2016 (Zhong et al., 2019). Here, we find the meridional difference between north and south is more obvious.*

Figure 3: Are these the average values?

The figure has been modified. The values are averaged in every period.

Lines 220-235, Figure 4: See one of the major points - this analysis does not consider that the gyre centre moves and area it covers expands/contracts over time. Given that the northern limit is only 79N, perhaps the stationary section is seeing a different part of the gyre/not capturing all of the northern extent in later years? You might see the same "equilibrium" if you took just the 73-76N range of "before 2008" plot, for example.

Thanks for this suggestion. We explained it and showed the modified figures above. Results in Fig. 4 are interpolated, so some observations at the northern and southern edges are missing. In period 1, the deepest point is only in the south (~74°N). However, there are similar deep points in the south (~74°N) and north (~77°N). We consider that the northern

gyre edge can reach 80°N. We rather focus on the part including the gyre centre and edge than just see the 73-76N range only near the gyre centre.

Line 256: "The cold-core anticyclones are popular in the BG region due to oceanic stratification and large-scale dominated circulation.". Why is this? Also, the word "popular" should not be used here – maybe "common"?

Thanks for pointing it out. We added the explanation.

> *(Line 264)*
>
> *The cold-core anticyclones are common in the BG region due to large-scale dominant anticyclonic circulation coupled with oceanic stratification, where cold and fresh Pacific water overlies warm and salty Atlantic water.*

Figure 5, section 4.1: There are some interesting features here. However, it would be nice to have a paragraph relating the individual eddy counts with the EKE profiles. For example, why does Mooring D have a similar profile of EKE in 2003-2007 and 2008-2014, but more eddies identified in 2008-2014 than 2003-2007? Does this mean that the deviation of velocities from the mean is contributing a lot to the EKE profile in 2008-2014?

Thanks for this suggestion, we added a paragraph relating eddy counts with the EKE profile.

> *(Line 293-300)*
>
> *At the southwestern corner (mooring A) of the basin, only 9 eddies were detected in the first period. EKE increased in the second period when there were 15 eddies and remained stable in the third period where were 13 eddies. Northwestern (mooring B) EKE was stronger with 14 eddies in the second period than before, despite 17 eddies detected in 2003–2007. And EKE was weaker in the third period due to less valid observations. Southeastern (mooring D) EKE did not occur apparent growth until the third period due to much stronger eddies detected. There were only 14 eddies in 2014–2018 and 24 eddies detected in 2008–2014. In short, there were either stronger eddies or much more eddies after 2008 than before.*

Lines 326, Figure 8: you have spent much of the paper describing the differences between the moorings (halocline properties and EKE). So you need to justify more why you are choosing to combine the mooring data here.

The explanation was added.

> *(Line 335)*
>
> *As shown in the eddy detection from MMP, eddies are common in the halocline layer. Results from MMP can well represent the variability in halocline eddies in the BG region, which are also consistent with former research. Results from every mooring*

*are thought equal to characterize the main features of eddy strength in the BG region, so EKE above the halocline base for different moorings are vertically averaged with depth to obtain the whole evolution over the years between 2003 and 2018.*

Lines 333-335: A fluctuation of both datasets doesn't seem to be the case between 2010 and 2015?

This paragraph was modified.

*(Line 341)*

*EKE from altimetry has increased gradually since the 1990s and peaked in 2009, and then, it decreased in 2009–2010, resulting in relatively weak and stable EKE in 2010–2015. Although the EKE from reanalysis is the highest estimate among them, it has also increased since the 1990s and remained at a stable level after 2010.*

Lines 339-341: Which datasets are you talking about here? MMP data seems to be higher since 2014. "Recently" should be specified, since oscillations occur at different times in each dataset.

This is talking about all EKE time series. This sentence was modified.

*(Line 349)*

*After experiencing a low ebb, especially from altimetry and MMP, since 2014/2015, EKE has presented some enhancement and oscillated around constant levels between the central BG and its marginal continental slope.*

Lines 334-335: which halocline variables? Do you mean depth and thickness from the first few figures in the paper? If so, refer to that here. The "plateauing" is only relevant for SODA and altimetry – MMP seems to decrease over this time period.

It means halocline depth and thickness. We corrected it.

*(Line 346)*

Between 2010 and 2015, EKE was relatively weak and even decreased in the two regions, lagging behind the plateauing of halocline depth and thickness.

Lines 357-358: it would help to guide the reader to the relevant part of the figure here (where the Alaska box is) as this is a new way of looking at the information.

We specified the location of the Alaska coast. And the Alaska box is marked in Figure 10.

Lines 359-362: This is the second time MKE is referred to. Since it is not shown, it should not be described as though it is referring to a figure unless it is of relevance to the discussion. What is the main point of talking about MKE here?

Thanks for pointing it out. The description of mean kinetic energy (MKE) was added. We are relating MKE and EKE here in this discussion. MKE is much smaller along Alaska region. EKE is dominant in kinetic energy.

Figure 10:  I would recommend putting the Alaska box on this map to help the reader.

The Alaska box was added.

[Figure]

Lines 390-391: see major point about asymmetry along the section

We explained it above.

lines 397-413: I found this paragraph hard to follow. It might help if figure was referred to more. Perhaps remind the reader what a positive value in A means, as you described in the methods. Why can't the salinity anomaly can be related to changes in freshwater rather than eddy transport?

Thanks for this recommendation. This paragraph was modified according to your suggestion.

> *(Line 418)*
>
> *In the first period, when the Eady timescale was relatively larger over the long term (Fig. 11c), meaning stronger stability, the salinity anomalies in the mixed layer and the halocline layer were both positive, more than 0.5 (Fig. 11b). Combined with the distribution pattern of the eddy stream function, the eddy thickness fluxes were generally positive at the surface, about 0.1 m²/s², and represented the southwards (northwards) propagation of low-salinity (high-salinity) water. ……*

Line 450: The proposed relationship between changes in the mixed layer and the tilt of the halocline should be explained much more clearly here.

Several instances:

- "abnormal" or "anormal" salinity should be clarified
- "mean time" is used a lot – is it meant to mean "average state"? Or "same time"?

This paragraph was modified according to your suggestion. The expression about salinity anomaly was unified. We checked the usage of "mean time" and distinguished two meanings. "mean time" is changed to"same time" or "average state" in the revised version. And some expressions are also modified in this discussion.

Reference:

Armitage, T., Manucharyan, G. E., Petty, A. A., Kwok, R., and Thompson, A. F.: Enhanced eddy activity in the Beaufort Gyre in response to sea ice loss. Nat. Commun., 11, 1-8. https://doi.org/10.1038/s41467-020-14449-z, 2020.Doddridge, E. W., Meneghello, G., Marshall, J., Scott, J., and Lique, C.: A Three-way balance in the Beaufort Gyre: The Ice-Ocean Governor, wind stress, and eddy diffusivity. J. Geophys. Res. Oceans, 124, 3107-3124. https://doi.org /10.1029/2018JC014897, 2019.

Kenigson, J. S., Gelderloos, R., and Manucharyan, G. E.: Vertical structure of the Beaufort Gyre halocline and the crucial role of the depth-dependent eddy diffusivity. J. Phys. Oceanogr., 51, 845-860. https://doi.org/10.1175/JPO-D-20-0077.1, 2021.

Manucharyan, G. E., and Spall, M. A.: Wind‐driven freshwater buildup and release in the Beaufort Gyre constrained by mesoscale eddies. Geophys. Res. Lett., 43, 273-282. https://doi.org/10.1002/2015GL065957, 2016.Manucharyan, G. E., Spall, M. A., and Thompson, A. F.: A theory of the wind-driven Beaufort Gyre variability. J. Phys. Oceanogr., 46, 3263-3278. https://doi.org/10.1175/JPO-D-16-0091.1, 2016.

Manucharyan, G. E., and Isachsen, P. E.: Critical role of continental slopes in halocline and eddy dynamics of the Ekman-driven Beaufort Gyre. J. Geophys. Res. Oceans, 124, 2679-2696. https://doi.org/10.1029/2018JC014624, 2019.

Manucharyan, G. E. and Stewart, A. L. (2022). Stirring of interior potential vorticity gradients as a formation mechanism for large subsurface-intensified eddies in the Beaufort Gyre. J. Phys. Oceanogr., 52, 3349-3370, https://doi.org/10.1175/JPO-D-21-0040.1, 2022.

Manucharyan, G. E., and Thompson, A. F.: Heavy footprints of upper-ocean eddies on weakened Arctic sea ice in marginal ice zones. Nat. Commun., 13, 1-10. https://doi.org/10.1038/s41467-022-29663-0, 2022.

Regan, H. C., Lique, C., and Armitage, T. W. K. : The Beaufort Gyre extent, shape, and location between 2003 and 2014 from Satellite observations. J. Geophys. Res. Oceans, 124, 844-862. https://doi.org/10.1029/2018JC014379, 2019.

Regan, H., Lique, C., Talandier, C., and Meneghello, G.: Response of total and eddy kinetic energy to the recent spin-up of the Beaufort Gyre. J. Phys. Oceanogr., 50, 575-594. https://doi.org/10.1175/JPO-D-19-0234.1, 2020.

Spall, M. A., Pickart, R. S., Fratantoni, P. S., and Plueddemann, A. J.: Western Arctic Shelfbreak eddies: Formation and Transport. J. Phys. Oceanogr., 38, 1644-1668. https://doi.org/10.1175/2007JPO3829.1, 2008.

Zhong, W., Steele, M., Zhang, J., and Cole, S. T.: Circulation of Pacific Winter Water in the western Arctic Ocean. J. Geophys. Res. Oceans, 124, 863-881. https://doi.org/10.1029/2018JC014604, 2019.

---

## Referee Report (RR1)

The manuscript has improved significantly from the previous version, both in writing and the overall flow. Although the authors have addressed most of my comments, some clarifications are still needed. I list those comments which require further clarifications below, together with some new thoughts.

[1] Halocline depths are used throughout the paper, in context with APE, Eady timescale, and EKE. These variables are more relevant to isopycnal steepness in the halocline (halocline strength as put in the manuscript), not the depth. Halocline depths are associated with both its steepness and water redistribution. It would be clearer if the authors discriminate these terms and use the right term in different sections/paragraphs.

[2] Paragraph 2-5 in the Introduction section: the authors listed many findings from previous literature, but these paragraphs lack a logical flow which connects the previous work. I recommend restructuring these paragraphs for clarity.

[3] Line 152: the authors focus on the PWW part of the halocline in the paper. However, the introduction reads like the entire halocline (from ~30m to ~250m) will be explored. It would be clearer if the authors can clarify their major research topic in the introduction.

[4] Line 240: Can the authors explain why a symmetrical shaped halocline implies a state of equilibrium? Isopycnal steepness can still change even after halocline becomes symmetric.

[5] Figure 5: Days of eddies are strongly related to how fast eddies were translated by background currents. This makes this metric less reliable to represent the eddy activities. Can the authors comment on this?

[6] Line 291: The mooring measurements go up to ~50 m so EKE is not surface-intensified. Section 5 implies that the authors define upper ~100 m as surface (which includes halocline). It would be better if names of different layers are defined in the beginning to avoid confusion.

[7] Figure 8b: The major topic of this section is interannual variability (seasonal cycle is only mentioned in one sentence). Showing monthly EKE makes it harder to identify the trend and interannual variability. I suggest showing a panel with annual mean EKE.

[8] Line 381: I'm still confused how the probability is estimated. Is it the frequency of days with eddies in a period?

[9] Line 419: mixed layer is only the upper ~30 m. As in my comment [6], please define terms in advance to avoid confusion.

[10] Line 432-433: Can the authors explain why low-salinity water is at the subsurface? With a bowl-shaped halocline, at the same depth, water at the edge should be saltier than that in the basin interior.

[11] I understand that surface EKE in AL/BSS regions are closely related to halocline eddies inside the basin, but I'm still not clear about the reason of analyzing surface EKE in basin

interior. The authors in their previous response explain that surface EKE also impacts halocline, but I don't find the consistency. I suggest the authors add a sentence or two to motivate their analyses of surface EKE in basin interior.

[12] I'm convinced with the authors conclusion about the role of eddies in modulating halocline shapes and depths. Besides, wind pattern is an important factor which can directly alter the center of the surface sea level dome and thus the area of Ekman pumping. It's important to at least discuss atmosphere's role in the discussion part.

---

## Author Response (AR2)

**Response to reviewers for manuscript "Long-term eddy modulation inhibited the meridional asymmetry of halocline in the Beaufort Gyre" by Lu et al.**

September 2023

The author comment is presented in the following sequence: (1) itemized comments from the reviewer in black, (2) author's response in blue, (3) quotations from the revised paper in indented *blue italic*.

**Anonymous reviewer #1:**

The manuscript has improved significantly from the previous version, both in writing and the overall flow. Although the authors have addressed most of my comments, some clarifications are still needed. I list those comments which require further clarifications below, together with some new thoughts.

Thank you very much for taking the time to review our manuscript. We agree on these constructive recommendations to clarify our scientific results. The manuscript is revised again according to these helpful recommendations.

[1] Halocline depths are used throughout the paper, in context with APE, Eady timescale, and EKE. These variables are more relevant to isopycnal steepness in the halocline (halocline strength as put in the manuscript), not the depth. Halocline depths are associated with both its steepness and water redistribution. It would be clearer if the authors discriminate these terms and use the right term in different sections/paragraphs.

Thanks for pointing it out. We carefully checked sections in the paper to distinguish these variables. The right terms of halocline depth and steepness are used in the revised paper according to your proposal. The variables like APE and isopycnal slope are relevant to halocline steepness. The deepening of halocline depth and its bowl-shaped structure is associated with halocline thickness, depth, and freshwater redistribution.

[2] Paragraph 2-5 in the Introduction section: the authors listed many findings from previous literature, but these paragraphs lack a logical flow which connects the previous work. I recommend restructuring these paragraphs for clarity.

Thanks for your recommendation. We have restructured these paragraphs to enhance clarity and coherence. We have now revised the introduction section by organizing the information in a more logical sequence. We have provided a concise summary of each previous finding and have highlighted how it relates to our study.

Paragraph 2 demonstrates halocline layer within Beaufort Gyre, especially on asymmetrical stratification. Paragraph 3 introduces eddy vertical distribution and horizontal patterns. Paragraph 4 retrospects previous literature about temporal variability of eddy activity under the background of variation in BG physics, like FWC and gyre strength. Paragraph 5 is about the role of eddies in the halocline dynamics.

[3] Line 152: the authors focus on the PWW part of the halocline in the paper. However, the introduction reads like the entire halocline (from ~30m to ~250m) will be explored. It would

be clearer if the authors can clarify their major research topic in the introduction.

Thanks for your recommendation. PWW is a main component of the halocline. To address this concern, we have revised the introduction to explicitly state that our research primarily focuses on the entire halocline. We emphasized the entire halocline that our study aims to examine.

[4] Line 240: Can the authors explain why a symmetrical shaped halocline implies a state of equilibrium? Isopycnal steepness can still change even after halocline becomes symmetric.

Thanks for pointing it out. "a state of equilibrium" is inappropriate here and it is replaced by "a state of stabilisation". The related sentences were modified in this paragraph.

In the previous period, the asymmetrical halocline was much steeper close to Beaufort sea southern slope and it was shallowing to the northern deep basin, which means stronger baroclinic instability in the south (Manucharyan et al., 2017; Manucharyan and Isachsen, 2019). The asymmetrical halocline influences that the halocline thickness in the south is much thicker than in the north. In the third period, the halocline structure was transformed to a symmetrical shape, with similar halocline depth and thickness surrounding the gyre center. The halocline is transformed to bowl-shaped with flatter isopycnals near the central gyre, which is consistent with the inflated halocline layer (Zhang et al. 2023), meaning a relatively stable state than before (Zhang et al. 2016). In this state, isopycnal steepness can still change with steeper halocline in the edge. This structure is directly driven by a surface Ekman convergence that creates a bowl-shaped halocline (Manucharyan and Spall, 2016).

> *(Line 261)*
> *······, the halocline depth and thickness tended to be meridionally symmetrical accompanied by flattened isopycnal slope surrounding central gyre, shaped like a horizontal bowl under the forcing of surface Ekman convergence (Manucharyan and Spall, 2016), indicating that it had reached a state of stabilisation (Zhang et al. 2016).*

[5] Figure 5: Days of eddies are strongly related to how fast eddies were translated by background currents. This makes this metric less reliable to represent the eddy activities. Can the authors comment on this?

Thanks for pointing it out. When we discerned days of eddies, the background currents, from velocity field without eddy activity, were removed. This metric is just to examine the eddy lifetime or duration every year to represent eddy activeness.

[6] Line 291: The mooring measurements go up to ~50 m so EKE is not surface-intensified. Section 5 implies that the authors define upper ~100 m as surface (which includes halocline). It would be better if names of different layers are defined in the beginning to avoid confusion.

Thanks for pointing it out. In section 2, the names of the mixed layer and halocline layer are defined. The related texts were modified.

> *(Line 143)*
> *To investigate the variation in the overall halocline and understand the shifting of oceanic stratification, we consider the depth of the potential density surface $\sigma$=27.4 (25) $kg \cdot m^{-3}$ to approximately represent the base (top) of the entire halocline layer (Timmermans et al., 2020). Accordingly, the depth of the surface mixed layer is also*

*identified by the halocline upper boundary (Bourgain and Gascard, 2011; Polyakov et al., 2018).*

[7] Figure 8b: The major topic of this section is interannual variability (seasonal cycle is only mentioned in one sentence). Showing monthly EKE makes it harder to identify the trend and interannual variability. I suggest showing a panel with annual mean EKE.

Thanks for this suggestion. Figure 8b with added annual mean EKE was modified accordingly.

[8] Line 381: I'm still confused how the probability is estimated. Is it the frequency of days with eddies in a period?

The probability is estimated by the frequency of area mean time series of EKE and geostrophic current in each period. We calculated the percentile of these values in every period to estimate here.

[9] Line 419: mixed layer is only the upper ~30 m. As in my comment [6], please define terms in advance to avoid confusion.

Thanks for pointing it out. The terms, including mixed layer and halocline layer, are checked thoroughly and defined. According to our definition, the depth of the surface mixed layer is less than 70 m in the central BG region.

> *(Line 174)*
> *The depth of the surface mixed layer is less than 70 m. The entire halocline layer underneath the mixed layer, including upper and lower halocline, is mainly at 70–250 m.*

[10] Line 432-433: Can the authors explain why low-salinity water is at the subsurface? With a bowl-shaped halocline, at the same depth, water at the edge should be saltier than that in the basin interior.

With a bowl-shaped halocline, water at the edge still be saltier than in the interior at the same depth. The occurrence of low-salinity water at the subsurface means local salinity decreased at that time, which contributed to deepened halocline depth and increased thickness, so the halocline structure was transformed.

[11] I understand that surface EKE in AL/BSS regions are closely related to halocline eddies inside the basin, but I'm still not clear about the reason of analyzing surface EKE in basin interior. The authors in their previous response explain that surface EKE also impacts halocline, but I don't find the consistency. I suggest the authors add a sentence or two to motivate their analyses of surface EKE in basin interior.

Thanks for pointing it out. We supplemented an explanation to clarify the consistency in the context.

> *(Line 359)*
> *Based on the previous works, surface eddy activities are directly responding to the extra wind energy input (Armitage et al., 2020), while the subsurface EKE is related to from baroclinic instability and APE release (Manucharyan and Spall, 2016;*

*Manucharyan et al., 2016). Eddy activities at the surface and subsurface are both linked with BG stabilisation, contributing to increased energy dissipation. ……*

[12] I'm convinced with the authors conclusion about the role of eddies in modulating halocline shapes and depths. Besides, wind pattern is an important factor which can directly alter the center of the surface sea level dome and thus the area of Ekman pumping. It's important to at least discuss atmosphere's role in the discussion part.

Thanks for this suggestion. The surface Ekman convergence related to wind pattern is a crucial role in the BG system (Manucharyan and Spall, 2016). We added the influence of atmospheric activity in the discussion part to make results more comprehensive and credible.

*(Line 491)*

*Many studies support that BG system is strongly affected by atmospheric dynamics that contribute to deeper halocline in the interior gyre. The center of the surface sea level dome and wind-forced Ekman pumping area are also highly sensitive to wind patterns (Manucharyan and Spall, 2016; Regan et al., 2019; Timmermans and Toole, 2023).*

**Anonymous reviewer #2:**

Second review of "Long-term eddy modulation inhibited the meridional asymmetry of halocline in the Beaufort Gyre" by Lu et al.

I appreciate the efforts the authors have gone to in order to address the reviews of the previous iteration, and I believe that the structure is now clearer than before. However, the manuscript still has some major problems that, in my opinion, make it unsuitable for publication in its current state. In particular, I do not believe that one of my major points has been adequately addressed.

I have outlined the main points below. Note that some of these have appeared since the previous review. This is either because of wording changes, or because the text was made clearer which made it possible to assess the scientific content without being as affected by difficult sentences. Note that there are still some places where the wording needs to be fixed.

Thank you for your valuable feedback on our paper. We are very grateful for your time and effort in reviewing our work. We have carefully considered your comments and have made these necessary revisions to address the concerns raised in responses to specific comments. The overall manuscript has been carefully checked and the incorrect wording was fixed.

**Major points**

**1) Gyre asymmetry**

I believe that the gyre symmetry found in the manuscript is not a valid interpretation of the data available. In my previous review, I questioned how the northward limit of the datasets (northernmost being 80N) affects the results. Specifically, if the gyre centre is more south, the halocline to the north of the centre – both within, and perhaps further north than the northern

extent of the gyre - can clearly be seen. If the centre is located more north along the section, though, then it is closer to the northern limit of the data and less information is available about what happens up to the northern limit of the gyre and beyond. It is important to acknowledge this as a large caveat, since the gyre has been shown in the literature to move northwards and westwards during some of the time period in question. The authors have provided an additional set of transects along 140W to demonstrate "similar shifts along these two sections". In my opinion, the section along 140W after 2014 demonstrates my point; when the gyre has moved westwards, the deepest part of it occupies less of the transect than it does along 150W, and there is a clear asymmetry in isopycnal slope either side of the deepest part, much like in 150W before 2008. I suggested that the authors could check if the northward limit is making them draw the conclusion that the gyre is now symmetric by analysing the SODA dataset further north than the data extends to. This has not been done.

Thanks for your suggestion sincerely. I have checked the impact of the BG northward limit in the results to improve scientific results. Analysing SODA and CTD, the annual meridional isopycnal slope (north to 90°N) in the halocline layer that represents halocline steepness was added in the supplement. Especially, the slope of the halocline base that varied remarkably within BG box in the first and third periods was added in Fig. 4c in the revised paper. Figure 11a with northmost halocline depth was accordingly modified northwards to 90°N.

Under the background of BG movements westwards and northwards (Regan et al. 2019; Bertosio et al., 2022), halocline variability and its asymmetrical structure are influenced. As shown in Figs. S1, the northward slopes of all isopycnal surfaces, with opposite slopes on the either side of gyre center show typical structure like a bowl under the forcing of surface Ekman convergence (Manucharyan and Spall, 2016). From the view of steeper isopycnals $\sigma$ =27.4 and 27, the steepest part of the halocline along this transect is occupied in the south of approximately 82°N (some part beyond BG region) over years 2004-2020. When the position of BG center is much closer to the southern slope (before 2008), the halocline lower boundary is steepest on the south side and it is shallowing from central gyre where isopycnal slope is near zero up to ~ 82°N (Fig. 11a), with meridional symmetry, which can be explained by the intrusion of Altantic water and strong Ekman downwelling in the central BG (Karcher et al., 2007; McLaughlin et al., 2004; Timmermans and Toole, 2023). To the further north of the BG region, the halocline is relatively flatter and its slope is negligible (Fig. 4c). When gyre location is to the north after 2014, the northern thickness increased more and thickness was similar surrounding gyre center. In the meantime, the main part of halocline where halocline is deepest and its slope is obvious is well captured in BG region. Using these available data including SODA and observations, although low-salinity anomaly exists in central Arctic just near 90°N (Fig. 11b), we find the variability of halocline structure and eddy activity in the area we focus on is much more remarkable than in the northmost extent. Though observation is just to about 80°, variability on the main part of halocline with time varying is still discerned within the fundamental BG region.

In previous manuscript, we provided an additional set of transects along 140°W to demonstrate there is a clear asymmetry in the isopycnal slope induced by deeper halocline in the south, much like in 150°W before 2008. The deepest parts of halocline along two

transects demonstrate that the BG movements influence the halocline. Comparing the views in the first and third periods, the similar deformation of halocline structure with halocline deepening more on the northern side of central gyre along two transect supports our results.
*(Line 238-243,249-255,258-264)*

I understand that much of the gyre is contained within the "BG box", and that it has been used in previous studies. However, there is a difference between computing integrated properties in a region (as is done in previous studies) and analysing vertical structure within different parts of the region. I believe that the latter is more heavily affected by the position of the centre and spatial variability, and so such analyses carried out in a static region must be very conservative in their conclusions. As it stands, in the later period it could equally be interpreted that only the central portion of the gyre is being shown. There is no evidence to demonstrate that this is not the case.

Thanks for pointing it out. We consider the spatial variability of BG (Regan et al., 2019), we put halocline structure to the north of BG box into comparison. For analysing vertical structure within different parts of the region, we compared isopycnal slope within and beyond BG region from 70°N to 90°N in the supplement. As interpreted in the upper response, halocline steepness in the further north is much smaller than in the south from SODA reanalysis. The variability on main part of BG halocline is well captured in the BG box. Considering the influence of the position of the centre and spatial variability on the halocline structure, we determined the position of gyre center by isopycnal slope of near zero in the interior of BG region. Results show gyre center moved and expanded northwards and the entire halocline inflated, corresponding with previous studies (Bertosio et al., 2022;Zhang et al., 2023). From the supplement, the halocline structure tended to be bowl-shaped with flatter halocline surrounding gyre center. Therefore, to the some extent movement and expansion of gyre center also influenced isopycnal slope on either side of central gyre and reduced the asymmetric structure of halocline, which demonstrates our results complementally.

The authors' response that the APE (and thus the halocline slope) flattens, and therefore the gyre becomes symmetric, is not so relevant as an argument, since it is computed within the same region as the section in question, and therefore does not contain information from further north.
I further note that Manucharyan and Isachsen (2019) suggest steeper isopycnal slopes in the south due to bathymetry; with a southern bound very close to the continental slope, the isopycnals will naturally be steeper there. Perhaps when the gyre centre is further from the continental slope, it allows for flatter isopycnals south of the centre away from the slope? This is not discussed.

Thanks for pointing it out. This inappropriate interpretation was modified in revised manuscript. We could not conclude the halocline becomes symmetric directly from APE evolution. APE within BG box, associated with eddy genesis and halocline strength, was analysed to explore BG stability here.
The influence of bathymetry in the south was added through previous literature. Here we take the position with isopycnal slope of near zero as gyre centre. With gyre centre further from the continental slope, partial isopycnals are flatter near BG center (Fig. 4c), while isopycnals

*are steeper on the edge due to enhanced eddy-induced convergence and BG strength.*

*(Line 249)*
*Initially, the location of central gyre that is determined by isopycnal slope of near zero in the interior, was very close to the vicinity of continental slopes with the largest isopycnal steepening occurring to the southern side and stronger baroclinic instability (Manucharya and Isachsen, 2019).*

The continued emphasis of the gyre becoming symmetric, which is attributed to eddies by the authors, without discussing how the changing shape/size/centre location of the gyre in itself affects the halocline structure, is a significant oversight. Being limited by the data range is not an excuse for this, especially since the authors could try to verify their conclusion is valid by looking at the SODA data.

Thanks for pointing it out. Sorry for the inappropriate texts related to asymmetry. These were modified. The main point in this study is asymmetrical halocline not gyre asymmetry. As shown in the current study (Regan et al., 2019), the shape of Beaufort Gyre is always asymmetrical. Now study on deformation of the halocline asymmetry is fewer. We just found that the asymmetry of halocline structure reduced in recent years. In the final period, it was more like a bowl shape. We understand changing shape, size, and centre location of the gyre all affect the halocline structure. We took the influence of BG into account but we did not carry out a detailed analysis, due to exploring how much impact on the halocline beyond the scope of this paper. As described above, according to isopycnal steepness in the halocline layer, the location of BG center and fundamental area of halocline were involved in our revised manuscript using SODA and observations in Section 3.2. The influence of gyre center location in halocline structure was specified in the revised manuscript. The eddy modulation in the oceanic stratification and changing gyre center location both influence halocline structure becoming symmetrical. We would discuss both of these factors in Section 6.

**2) Misleading description of figures**
In a number of places, descriptions of figures do not match what the figures show. These ambiguities, and in some cases errors, make it hard to know whether the text is correct or not. I list some examples below:

Figure 2 and Table 1, and associated text (section 3.1) is very confusing. There are multiple places where statements in the text are not apparent in the figure and not backed up by Table 1. Firstly, there seems to be contradictions related to whether a negative trend in halocline depth is a decrease or increase in depth. "A negative trend of halocline depth is clear during 2008-14 in the southern sites of the basin (moorings A and D), but the former and latter periods both mostly exhibit positive trends in halocline depth and thickness" would suggest that positive is a shallowing depth, and decrease in thickness. But then, for mooring B, "the halocline depth continues deepening over the whole period" which has positive trends for each time period, suggesting positive is deepening. Likewise, in the response to review, "the negative trend means halocline depth is lifting during that period, which is not comparable with the deepening trend" suggests positive is deepening. Such inconsistencies make it hard to follow what is actually happening and when, especially since the general trends (increasing

or decreasing) are hard to tell in figures 2a and 2b due to the y-axis and short-term variability.
Thanks for pointing it out. These associated texts were checked for better clarification. A negative trend means a decrease in depth and a decrease in thickness. For better clarification, the original "negative trend" was replaced by "shallowing trend". Likewise, "positive trend" was replaced by "deepening trend". For comparison, the deepening trends of halocline base over the whole period were added in Fig. 2a. At four moorings, the depth halocline base are all exhibiting long-term deepening trends over the whole period 2003-2018, which means deepening in halocline depth.

Figure 2c also has poor text accompanying it. Lines 200-202 do not correctly analyse the figure - there is not a continuous increase before 2009, there is not a continuous decrease between 2010 and 2014 (it goes up between 2010 and 2011!), and it has not been relatively stable since 2015 (the increase from 2017-2018, for example, appears similar to the decrease between 2011 and 2014). This also appears in line 400.
Thanks for pointing it out. There was an error of x-axis in Fig. 2c. The text related with Figure 2c was revised accordingly.
> *(Line 203)*
> *As shown in Fig. 2c, there was a significant increase from ~50 to ~200 PJ before 2010. However, APE was continuously decreasing in the periods of 2010–2013, 2014–2016 and 2017–2021, meaning a flattening of isopycnals in the BG and APE release. Although there was a short-term increase in APE accumulation in 2016-2017, it did not lead to further accumulation, indicating an energy reservoir limit around 150-250 PJ.*

Figure 3 (line 220): after 2014, there is also a difference north-south (just with some fresher water in the far south). So I do not really see a gradual decline in spatial difference
Thanks for pointing it out. "a gradual decline in spatial difference" is not appropriate here. Associated texts were revised.
> *(Line 223)*
> *In the initial period, the halocline base maps highlight significant differences between the north and south. And then the north experiences a much more pronounced deepening of the halocline depth compared to the south closer to the Beaufort Sea slope, which also exhibits more fresher water.*

Figure 4: what is shown by the arrows? The caption says "the depths of halocline base on either side are marked" – either side of what? Is it the interannual mean of the centre? I worry that the two points are not comparing the same parts of the gyre on each plot
In Fig. 4a, we added the interannual mean centre position determined by isopycnal slope of near zero in the interior. The arrows were modified, showing the depth of halocline base on either side of central gyre.

Figure 6, and lines 291-293: many lines on the figure, but most notably the green lines for moorings a and d, are neither bimodal nor decaying directly from the surface.
Thanks for pointing it out. This sentence was modified.

*(Line 310)*

*The vertical structures of EKE in the basin and its marginal seas can be classified into two types. The first type is that EKE is up to ~ 0.01 m2/s2 under the surface mixed layer and decays with depth. The second type is with maximum value at the subsurface of approximately 70–250 m between the upper and lower halocline boundaries.*

Figure 7: you state on line 318 that "EKE was significantly enhanced compared with that in the previous period" – are you just talking about the area in the Alaska box? This should be stated here. There are parts where EKE clearly decreases over time in the rest of the figure.

Thanks for pointing it out. This sentence was modified.

*(Line 337)*

*In every period, the EKE field along the southern continental slope was significantly enhanced compared with that in the previous period, and the strong EKE gradually developed from coasts to offshore regions and the central basin with time.*

Figure 8: Lines 340-355 say some incorrect things about the figure. For example, line 345 suggests SODA has been stable since 2010, but in the figure it reaches its highest points during strong variability from 2014 onwards, and this variability is comparable to the increase from the mid 1990s to 2010. Line 347 states EKE decreased in the two regions between 2010 and 2015, but it seems to increase in both SODA and altimetry, and in the case of SODA it is not "relatively weak" – it is higher than in all previous years before 2009.

Thanks for pointing it out. The associated texts were corrected.

*(Line 366-374)*

*Although the EKE from reanalysis is the highest estimate among them, it has also increased since the 1990s and remained at a relatively stable level in 2010–2013. In the BG region, subsurface EKE began to increase rapidly since 2003 and peaked in 2009, and it indicated a decrease until 2014, which was slightly different from that in the AL region. When EKE was relatively strong after increasing, its cumulative effect was contributing to plateauing of halocline depth and weakening of halocline steepness associated with APE release. These characteristic shifts of eddy and oceanic stratification were both related to the varying physics of the gyre in the upper layer that indicated a strengthening during the years before 2007 and a possible stabilisation since 2008 (Zhang et al., 2016). After experiencing a low ebb, especially from altimetry and MMP, since 2014/2015, EKE has presented some enhancements over time and remained at higher levels than in previous years before 2008 between the central BG and marginal AL regions.*

**Response to other revisions**

You discuss MKE in multiple places, and I asked why it was relevant. It would be useful to add another set of subplots showing the MKE, since it seems that the authors find the comparison with EKE important.

Thanks for this suggestion. A subplot of MKE was added in Fig. 7.

One of my comments asked for further justification that you are combining the mooring data, after focusing on how they are different. The new lines 335-339, which address this, should have a reference about which particular former research is relevant here. Additionally, "every mooring are thought equal to characterize the main features of eddy strength in the BG region" is not in itself a justification – do you have a reference for this? Combining the moorings and vertically averaging should be explained more here, in particular in how it affects the results. For example, 2013 and 2014 have very low EKE, but mainly have data from moorings A and D in them – is this a coincidence?

Thanks for this suggestion. The interpretation of combining mooring observation was added. From observations, the study on the long-term variability of EKE is few. The variability of eddy counts from MMP can well represent the variability in halocline eddies, for example, more eddies are detected in the southwestern sector of Canada Basin (Mooring A) in 2005, 2009 and 2013, which is well consistent with former research by ITP observations (Zhao et al., 2016). The results is verified in other sectors. Therefore, considering the difference in spatial distribution of eddy activity, we use horizontal velocities of all moored observations to quantify eddy strength. As shown by previous literature, eddy activity is most active in the southwestern sector of CB (Mooring A). Very low EKE in 2013 and 2014 can be supported by relatively fewer eddy days related to eddy duration.

> *(Line 355)*
> *As shown in the eddy detection from MMP, eddies are common in the upper and lower halocline of the CB (Zhao et al., 2014). The variability of eddy counts in four moorings are also consistent with former research by ITP observations in four sectors of the CB (Zhao et al., 2016), so EKE above the halocline base for different moorings is vertically averaged with depth to comprehensively characterize the main features of eddy strength over the years between 2003 and 2018.*

**Other comments**

Lines 63-67: time periods of changes of each variable should be provided, and the literature checked to ensure it is correct. For example, I believe Zhang et al (2016) discuss a stabilization rather than continued increase in some properties. The phrasing as it stands could suggest a continuous increase in all gyre properties over the last decades.

Thanks for this suggestion. We checked the related literature.

Line 236, 238: Be careful when talking about "lower" depth. Here it seems that you are using lower as deeper, but when you say "30m lower" it could mean "30m less". Prefer to use "deeper" or "shallower" to describe depth changes

Thanks for pointing it out. We checked related texts and used "deeper" or "shallower" to describe depth changes according to your recommendation.

Line 299: what is meant by "less valid observations"?

"less valid observations" means in third period there are fewer observations at mooring B (Fig. 5, thin grey lines). It was modified.

"before 2008" seems to mean different time periods in different parts of the manuscript. This

should be clarified everywhere, as otherwise it is hard to compare figures.

The years of "before 2008" were clarified in every figure for better comparison.

Line 347: it is strange to phrase a sentence as EKE lagging behind halocline changes, since I thought the point was to show how EKE was causing halocline changes. This should be clarified.

Thanks for your suggestion. This sentence was modified for explaining relationship between EKE and halocline.

> *(Line 362)*
> *When EKE was relatively strong after increasing, its cumulative effect was contributing to plateauing of halocline depth and weakening of halocline steepness associated with APE release.*

**References**

Bertosio, C., Provost, C., Athanase, M., Sennéchael, N., Garric, G., Lellouche, J. M., Bricaud, C., Kim, J. H., Cho, K. H., and Park, T.: Changes in freshwater distribution and pathways in the Arctic Ocean since 2007 in the mercator ocean global operational system. J. Geophys. Res. Oceans, 127, e2021JC017701. https://doi.org/10.1029/2021JC017701, 2022.

Karcher, M., Kauker, F., Gerdes, R., Hunke, E. and Zhang, J: On the dynamics of Atlantic Water circulation in the Arctic Ocean. Journal of Geophysical Research, 112, C04S02, https://doi.org/10.1029/2006JC003630 , 2007

Manucharyan, G. E., and Spall, M. A.: Wind-driven freshwater buildup and release in the Beaufort Gyre constrained by mesoscale eddies. Geophys. Res. Lett., 43, 273-282. https://doi.org/10.1002/2015GL065957, 2016.

Manucharyan, G. E., Thompson, A. F., and Spall, M. A.: Eddy memory mode of multidecadal variability in residual-mean ocean circulations with application to the Beaufort Gyre. J. Phys. Oceanogr., 47, 855-866. https://doi.org/10.1175/JPO-D-16-0194.1, 2017.

Manucharyan, G. E., and Isachsen, P. E.: Critical role of continental slopes in halocline and eddy dynamics of the Ekman-driven Beaufort Gyre. J. Geophys. Res. Oceans, 124, 2679-2696. https://doi.org/10.1029/2018JC014624, 2019.

McLaughlin, F. A., Carmack, E. C., Macdonald, R. W., Melling, H., Swift, J. H., Wheeler, P. A., Sherr, B. F. and Sherr, E. B.: The joint roles of Pacific and Atlantic-origin waters in the Canada Basin, 1997–1998. Deep-sea Res., 51, 107-128. https://doi.org/10.1016/j.dsr.2003.09.010, 2004.

Regan, H. C., Lique, C., and Armitage, T. W. K. : The Beaufort Gyre extent, shape, and location between 2003 and 2014 from Satellite observations. J. Geophys. Res. Oceans, 124, 844-862. https://doi.org/10.1029/2018JC014379, 2019.

Zhang, J., Steele, M., Runciman, K., Dewey, S., Morison, J., Lee, C., Rainville, L., Cole, S., Krishfield, R., Timmermans, M., and Toole, J.: The Beaufort Gyre intensification and stabilization: A model-observation synthesis, J. Geophys. Res. Oceans, 121, 7933-7952. https://doi.org/10.1002/2016JC012196, 2016.

Zhang, J., Cheng, W., Steele, M., and Weijer, W.: Asymmetrically stratified Beaufort Gyre: mean state and response to decadal forcing. Geophys. Res. Lett., 50. e2022GL100457.

https://doi.org/10.1029/2022GL100457, 2023.

Zhao, M., Timmermans, M., Cole, S., Krishfield, R., and Toole, J.: Evolution of the eddy field in the Arctic Ocean's Canada Basin, 2005-2015. Geophys. Res. Lett., 43, 8106-8114. https://doi.org/10.1002/2016GL069671, 2016.

---

## Author Response (AR3)

**Response to reviewer #2 and Editor for manuscript "Long-term eddy modulation affected the meridional asymmetry of halocline in the Beaufort Gyre" by Lu et al.**

October 2023

Responses, point-to-point, are given below in blue.

**Anonymous reviewer #2:**

The manuscript text has clearly improved from the last iteration, and I appreciate the efforts the authors have put in to address my previous concerns. In particular, I am pleased that the authors took on board my comments about the spatial extent of the gyre and provided more evidence to support their claims. I still have some further comments, however, which I have outlined below.

Note that while the authors have demonstrated that there is more eddy activity in the region, I still do not feel that the gyre positioning has been emphasised enough as an additional potential contributor to the halocline changes described. While in essence this is just a case of wording, I feel it is necessary to state these potential caveats at the relevant points in the text. I think the main issue I have is that the title directly attributes the changes to eddies, which does not leave room for any other factors.

Thank you very much for taking the time to review our manuscript again. These suggestions were adopted in our revised paper where we believe they were useful to impress our thoughts. We have carefully considered your further comments and made corresponding modifications. While in this paper only eddy modulation is analysed, eddy is not the only factor in the halocline variability. Especially, we modified the title as "Long-term eddy modulation affected the meridional asymmetry of halocline in the Beaufort Gyre".

**Abstract**

Line 16, "reduced in the final" – the end of this sentence is missing.

Thanks for pointing it out. The wording was corrected.

Lines 18-19: "halocline structures on either side of the central gyre reached a nearly identical and stable regime" – how far either side is being considered? It has already been acknowledged in the first period the gyre was closer to the continental slope and so was affected by that. What distance is considered "close" when considering asymmetry?

Thanks for your recommendation. The distance from the central gyre was considered. Considering in different periods the position of central gyre moved, here we just took the position in the middle of either edge and central gyre as "either side" that is at least 120 km from gyre center.

**Introduction**

I feel like the structure of the introduction is still a little odd. From lines 48 onwards, we are first told about previous studies about eddies without knowing why it is relevant. We are then told the focus of the paper. We are then told about importance of eddies, before reading a paragraph that mixed the three topics. I found the logical progression and the role of each paragraph a bit unclear.

Thanks for this suggestion. We modified the structure of the introduction to be clearer. The importance of eddies and its impacts on the halocline are introduced first before listing previous studies about eddies. The sequence of some sentences is appropriately modified for a better flow.

Line 36: what happened to freshwater content after 2012?
"While in 2013 FWC decreased, it again increased from 2014." was added here.

Lines 44-45: why are isopycnals steeper nearer the gyre edge due to topography? In the north this is not the case.
Because southern gyre edge was limited by topography, eddy diffusivity increases towards coastal gyre edge. Eddies can redistribute isopycnal layer thicknesses laterally and as a result affect the halocline depth (Manucharyan et al., 2016). In the north there is an abyssal plain.

**Data and methods**
Line 98: why do the reanalysis datasets "mainly" consist of the two listed? What else is there?
"mainly" was removed. There are no else reanalysis datasets.

Lines 106-108: If the shallowest moorings start from between 50 and 90 metres, how does this affect the computation of the top of the halocline/thickness in Figure 2a/b and Table 1? The diagnosed "top" is consistently between 50 and 90 metres, which would suggest that it is above the first available data points sometimes. How does this affect the results?
Thanks for pointing it out. Considering there was no available moored data above 50 m, we used CTD observations to diagnose the depth of halocline top and halocline thickness. However, there are only annual CTD surveys and not enough sample counts, so trends of halocline top and halocline thickness were not calculated by CTD. Figure 2 and Table 1 were modified accordingly.

Lines 124-126: You state that the dynamic topography dataset is available in ice-free regions. What do you do when there is ice? You may be comparing quite different domain sizes when computing the EKE – does that affect the interpretation of the results?
Considering surface eddy activity is strongest during ice-free seasons (e.g., Meneghello et al, 2021), we optimally interpolated long-term EKE daily time series from altimetry data to supplement the missing value in ice seasons. Based on interpolated data, we calculated annual mean values. When we compare EKE variability in different domains, the results are averaged in the area.

**BG halocline variability**

Lines 191-193: I don't think you can say the variations at both northern sites cover 3 periods. Mooring C is only available for the first period. So, it is only Mooring B that is relevant here. Results from CTD observations were supplemented to address the missing of mooring data. The related texts were modified accordingly.

Lines 216-219: I think you should state what Zhong et al (2019) find explicitly here. What rates did they find? I would suggest that if they also find more deepening in the northwestern basin, then your findings do not "deviate" from previous findings. Thanks for this suggestion. We checked these sentences and made necessary modification.

Line 240: re-iterate that you are talking about the supplementary figure here. You also need to state explicitly why you have introduced a supplementary figure before you reference it (not just at the end of the paragraph) – just state that you are verifying that the behaviour north of the extent of the dataset is consistent. Thanks for this suggestion. We added the statement here.

Lines 259-265: As in the abstract, be very clear about what you mean by "close to gyre center". How far away do you need to be to still be classed as close? It looks like Figure 4c shows a steeper north close to the gyre after 2014… (around 77N). In general, I still think it needs to be explicitly said that **a)** the fact that the gyre has moved northwards means its southern edge is not as affected by the topography (and resulting asymmetry), and **b)** the section is not always capturing the same portion of the gyre. I know the latter point was meant to be demonstrated by taking a section along 140W, but to me this demonstrates more the thickening than the asymmetry – it shows that there is variability based on what specific section you take at a given time, and that the gyre moves. Figure 3 shows the part of the gyre that is being captured – after 2014, it is more the centre than before. Thanks for this suggestion. The inappropriate interpretation was modified. After 2014, there was steeper halocline on the northern side. This was added (Line 263). We emphasised the effect of gyre northward movement, which induced less affect by the southern slope (Line 262). I agree that the section is not always capturing the same portion of the gyre. The effect of the variability in the BG spatial movements on halocline via different sections and more areas of gyre center in the final period were added accordingly (Line 230).

**Spatiotemporal variability in eddy activity**
Line 363: How much is this affected by presumed less open water periods before 2003? Some points before 2003 are of a similar magnitude to points after 2003, is it ok to rule them out? The inappropriate interpretation was modified. There was low APE within BG region initially during 2003-2005 and more missing velocity data before 2003. Because EKE is strongest in the AL region, some points before 2003 are of a similar magnitude to those after 2003. From the views of surface EKE patterns, EKE was relatively low in the first period. Therefore, we do not emphasize eddy variability during this period here.

Line 373: Can you provide a summary sentence about the main takeaway message from the

three datasets over time? There is a lot of detail here but hard to know what the overall conclusion is.

Thanks for your suggestion. The related texts were added.

**Eddy modulation in the asymmetrical halocline**

Lines 423-424: which EKE plot are we referring to here? A figure reference would be useful.

Thanks for pointing it out. It was added.

Line 429: Is it a problem that the error of 30-40 metres is of the same order of the deepening seen in the data?

The error of 30-40 m is existing along the whole transect and over three periods. Despite of an overestimate of halocline depth, the deformation of halocline structure is captured.

Line 493: You cannot use the word "credible" here.

It was removed.

**Editor:**

I have read the revised paper myself and agree with the reviewer that there are still some areas requiring clarification. The attached pdf has some highlights where wording needs to be rephrased or clarified. Perhaps most importantly, I did not find the paper clear about what was shown that was new. You have obviously done a great deal of analysis and work, and you present plenty of nice figures and numbers. In many places you set this in the context of the literature. That's all good. But the summary and discussion section states again the things you have done and the results found. It would help if you could specifically and briefly identify what was new about the findings (i.e. had not been stated in the literature previously) and why this matters? e.g. This disagrees with..... or This extends the analysis of .....into a different region/time? You state in the Conclusions "This paper provides a possible perspective..." - can you state what that perspective is exactly? I was not sure what you meant here.

Please revise the paper taking into account the concerns of the reviewer and my suggestions above. If I consider that you have suitably addressed the reviewer's concerns, it should not be necessary to undergo further external review.

Thank you for your prompt and constructive comments on my manuscript. I have carefully reviewed the attached PDF to improve the manuscript further. I acknowledge the comments regarding the need for more clarification in certain areas of the paper. The wording was rephrased and clarified where necessary, particularly in the summary and discussion section. Regarding the suggestion of identifying what was new about the findings and why it matters, I rephrased the key contributions of the paper in section 6. I revised the statement in the conclusion, in a briefer manner, to more precisely identify the perspective offered by the paper.

---

## Author Response (AR4)

**Response to Editor for manuscript "Long-term eddy modulation affected the meridional asymmetry of halocline in the Beaufort Gyre" by Lu et al.**

October 2023

Responses, point-to-point, are given below in blue.

**Editor:**

There are a few odd wordings, which I hope will be picked up in the copy-editing process, but when uploading your final versions I would particularly like you to change the word 'transportation' to 'transport'.

Line 299 transportation should be changed to transport

Line 459 transportation should be changed to transport

Line 449, southwards transportation should be changed to southward transport.

Line 457 transportation should be changed to transport

Line 499 northwards transportation should be changed to northward transport.

Thanks for pointing it out. We checked the full paper. The words are all changed according to your advice.